# Modality-Specialized Synergizers for Interleaved Vision-Language Generalists

**Zhiyang Xu**[*1]    **Minqian Liu** [*1]    **Ying Shen**[1]    **Joy Rimchala**[2]
**Jiaxin Zhang**[2]    **Qifan Wang**[3]    **Yu Cheng**[4]    **Lifu Huang**[1,5]
[1]Virginia Tech    [2]Intuit AI Research    [3]Meta AI    [4]The Chinese University of Hong Kong
[5]University of California, Davis
{zhiyangx, minqianliu}@vt.edu, lfuhuang@ucdavis.edu

## Abstract

Recent advancements in Vision-Language Models (VLMs) have led to the emergence of Vision-Language Generalists (VLGs) capable of understanding and generating both text and images. However, seamlessly generating an arbitrary sequence of text and images remains a challenging task for the current VLGs. One primary limitation lies in applying a unified architecture and the same set of parameters to simultaneously model discrete text tokens and continuous image features. Recent works attempt to tackle this fundamental problem by introducing modality-aware expert models. However, they employ identical architectures to process both text and images, disregarding the intrinsic inductive biases in these two modalities. In this work, we introduce MODALITY-SPECIALIZED SYNERGIZERS (MoSS), a novel design that efficiently optimizes existing unified architectures of VLGs with modality-specialized adaptation layers, i.e., a Convolutional LoRA for modeling the local priors of image patches and a Linear LoRA for processing sequential text. This design enables more effective modeling of modality-specific features while maintaining the strong cross-modal integration gained from pretraining. In addition, to improve the instruction-following capability on interleaved text-and-image generation, we introduce LEAFINSTRUCT, the first open-sourced interleaved instruction tuning dataset comprising 184,982 high-quality instances on more than 10 diverse domains. Extensive experiments show that VLGs integrated with MoSS achieve state-of-the-art performance, significantly surpassing baseline VLGs in complex interleaved generation tasks. Furthermore, our method exhibits strong generalizability on different VLGs.[1]

## 1 Introduction

As multimodal learning research advances, there is a growing trend of building Vision-Language Generalists (VLGs) (Sun et al., 2023b; 2024; Koh et al., 2023; Aghajanyan et al., 2022; Li et al., 2023b; Dong et al., 2024; Team, 2024) that can comprehend and generate *interleaved* text and images, where multiple text segments and images are presented in arbitrary sequences. Compared with previous Vision-Language Models (VLMs) (Alayrac et al., 2022; Li et al., 2023c; Liu et al., 2023c; Fu et al., 2024; Qi et al., 2024b) that can only generate text and diffusion models (Ramesh et al., 2021; Rombach et al., 2022) that can only produce images, such VLGs enable a wider array of applications that require the simultaneous generation of both images and text, such as script generation (Qi et al., 2024a), visual storytelling (Huang et al., 2016), and many others.

Despite these recent advancements, one notable issue of existing VLGs is that they often fail to produce coherent and high-quality interleaved text and images. As shown in the top example in Figure 1, current state-of-the-art VLG, e.g., Emu2 (Sun et al., 2024), still suffers from poor text and image quality, including heavy repetition in text and unnatural distortions in the image. We attribute this issue to a fundamental challenge: *existing VLGs use the same architecture (i.e., transformer back-*

---

[*]Zhiyang Xu and Minqian Liu contributed equally to this work.

[1]We publicly released the code, model checkpoints, and dataset at `https://github.com/VT-NLP/MoSS`.

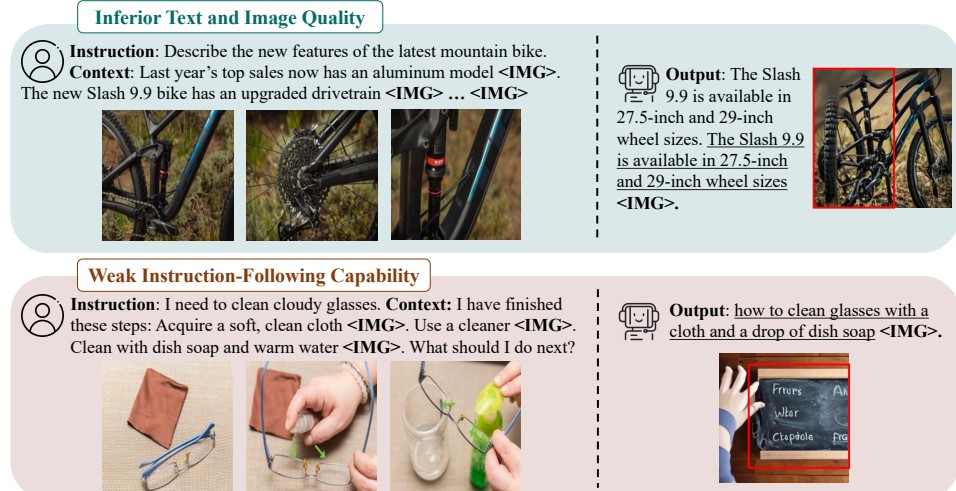

Figure 1: Failure cases of existing VLGs (Emu2 at the top and GILL at the bottom). The output text with inferior quality is highlighted with underline. The regions that impede output images' quality are highlighted with red bounding boxes.

*bone) with the same set of parameters to process both text and images, which may not be sufficient to model the distinct inductive biases in each modality given their intrinsic discrepancy.* For example, text follows a linear, left-to-right sequence, whereas images are inherently two-dimensional, composed of local priors in adjacent patches. Previous studies show that the transformer architecture (Vaswani et al., 2017) predominantly employed in current LLMs and VLGs excels at sequence modeling. But compared with convolutional architectures, transformer is less effective at modeling local priors of adjacent image patches, which is crucial for various vision tasks (Zhong et al., 2024; Chen et al., 2023d). Thus, applying a unified architecture with the same set of parameters can result in poorer performance in mixed-modal generation, such as producing images with local inconsistency and distortion among adjacent patches. Recently, several works (Akbari et al., 2023; Ye et al., 2024) have proposed modality-aware expert models as an attempt to tackle this problem. However, they still apply the same architecture to process both text and images, ignoring the inherent discrepancies between these two modalities. These challenges underscore the need for careful architectural design and specialized allocation of model parameters tailored to each modality.

Another critical challenge is existing VLGs (Sun et al., 2023b; 2024) often fail to adhere to human instructions to perform interleaved generation tasks. In the bottom example in Figure 1, GILL (Koh et al., 2023) failed to accurately follow the instruction and the context to complete the next step for *"clear cloudy glasses"*. Instead, GILL produces unhelpful text that is repetitive with the input and an image irrelevant to the task. While existing VLGs are often pretrained on interleaved documents (Zhu et al., 2023c), they are only instruction-tuned for single-modality generation, e.g., either text or image generation, leading to a weak instruction-following capability of interleaved generation. Moreover, there is a lack of large-scale instruction-following data specifically designed for interleaved generation, making the interleaved instruction tuning not scalable and less feasible.

To address these fundamental challenges, we first propose MODALITY-SPECIALIZED SYNERGIZERS (MoSS), a novel framework that introduces modality-specialized parameters to seamlessly handle the inductive biases of different modalities within the unified architectures of VLGs. As lightweight adaptation layers, our proposed MoSS is generic and can be integrated into most, if not all, existing VLGs without requiring expensive pre-training. Specifically, for images, we introduce **Convolutional** Low-Rank Adaptation (Convolutional LoRA) layers to better model the local prior of image patches. For text, we employ a separate set of **Linear** Low-Rank Adaptation (LoRA) layers, acknowledging the distinct sequential modeling process of text compared to images. During finetuning, both modality-specialized architectures are zero-initialized and progressively fine-tuned to learn their modality-specific features, while the VLG's parameters remain frozen, maintaining strong cross-modal integration gained from massive pre-training. Our design allows each modality to have a better representation of modality-specific features with its own specialized parameters and optimal architectural design. Additionally, to improve VLG's instruction-following capabilities under diverse interleaved generation scenarios, we introduce LEAFINSTRUCT, the first open-sourced

high-quality interleaved instruction tuning data with 184,982 instances spanning more than 10 domains. To obtain high-quality instruction data at scale, we develop a rigorous automatic pipeline.

To validate the effectiveness and generalizability of our method and dataset, we adopt our method on two different VLG backbones with discrete and continuous image token spaces, and conduct extensive experiments on multiple datasets. The results demonstrate that the VLGs instruction-tuned with our method achieves state-of-the-art performance across most evaluation aspects. Particularly, our method can produce interleaved content with better quality, including text quality, image coherence, text-image consistency, and helpfulness. In summary, our contributions are threefold. **First**, we introduce MOSS, a novel design that enhances VLGs to generate interleaved content with modality-specialized parameters and adaptation architectures. To the best of our knowledge, we are the first to apply different adaptation architectures within an autoregressive generative model to improve interleaved generation. **Second**, to fill the blank in existing resources and improve the instruction-following capability of VLGs, we introduce the first open-sourced large-scale instruction-tuning dataset across diverse domains. **Third**, by instruction-tuning existing VLGs with a small number of parameters, we achieve significant performance improvement on most aspects of evaluation benchmarks, outperforming existing open-source baselines by 34.7% on InterleavedBench. We also demonstrate that our approach can effectively generalize to different VLG backbones.

## 2 RELATED WORK

**Interleaved Vision-Language Models** There are two popular formulations for VLGs: The first leverages VQGAN (Esser et al., 2021) to quantize an image into a long sequence of discrete tokens and add the vocabulary in VQGAN's codebook into the vocabulary of LLMs (Aghajanyan et al., 2022; Yu et al., 2023; Yasunaga et al., 2023; Team, 2024; Jin et al., 2023). In this way, the LLMs are trained with a unified autoregressive objective to predict image tokens or text tokens. The predicted image tokens are fed into a VQGAN decoder to reconstruct images. The second formulation employs the CLIP image encoder to transform images into sequences of continuous embeddings (Koh et al., 2023; Tang et al., 2023; Zhu et al., 2023b; Sun et al., 2023b; 2024; Li et al., 2024b; Wu et al., 2023; Tian et al., 2024), which are then concatenated with text embeddings in their original order. Compared to the first approach, this formulation often requires shorter sequences to represent an image and generally yields superior performance. Our proposed method requires minimal assumptions on VLG's architectures and can be applied to many of the existing transformer-based VLGs.

**Visual Instruction Tuning** Xu et al. (2023) propose MultiInstruct, the first human-label visual instruction tuning dataset to improve the generalizability of VLMs. LLaVA (Liu et al., 2023c) leverages GPT-4 to convert image captions from existing annotations into three tasks, including visual dialogues, visual question answering, and detail captions. Following studies either utilize proprietary LLMs (Dai et al., 2023; Ye et al., 2023; Yin et al., 2023; Liu et al., 2023b; Li et al., 2023a; Lyu et al., 2023; Zhu et al., 2023a; Wang et al., 2023; Chen et al., 2023b; Zhang et al., 2024) or human efforts (Liu et al., 2023b; Xu et al., 2024) to augment visual instruction tuning tasks. Several studies target specific aspects of VLMs' capability, such as domain and instruction bias (Avrahami et al., 2022; Liu et al., 2023a), object grounding (Chen et al., 2023a), and OCR (Zhang et al., 2023b; Hu et al., 2023). Instruction tuning has also been widely applied to other vision-language tasks, such as image editing (Brooks et al., 2023a) and interleaved text-image understanding (Jiang et al., 2024). Hu et al. (2024) finetune a model that can follow multimodal instructions to generate desired images. However, most existing instruction-tuning datasets only consider the tasks where the outputs are in a single modality, i.e., either text or image. *To facilitate the training and enhance the instruction-following capabilities for VLGs, we curated* LEAFINSTRUCT*, the first instruction-tuning dataset tailored for interleaved text-image generation across diverse domains, where the inputs and outputs can contain interleaved text and multiple images.*

**Parameter-Efficient Finetuning (PEFT)** PEFT methods (Hu et al., 2021; Li & Liang, 2021; Karimi Mahabadi et al., 2021; Zaken et al., 2022; Jia et al., 2022; Lian et al., 2022; Jie & Deng, 2022; Liu & Huang, 2023; Liu et al., 2022; Chen et al., 2023d; Zhong et al., 2024) aim to adapt pretrained large models to various downstream tasks and have become prevalent in instruction tuning. Typically, these methods involve freezing the pretrained large models while finetuning a minimal set of newly introduced parameters. Recent studies (Wang et al., 2022; Zadouri et al., 2023; Lin et al., 2024; Shen et al., 2024) propose to combine PEFT methods with Mixture-of-Experts to mitigate task interference and enhance performance, particularly in visual instruction tuning where models need to process inputs from two modalities. *Our proposed* MOSS *is the first PEFT method that*

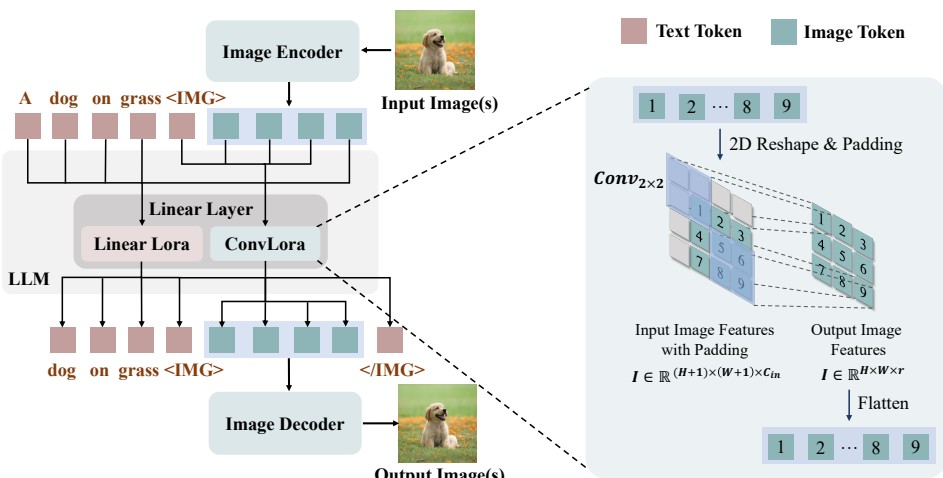

Figure 2: An autoregressive VLG with our proposed MoSS added to its linear layers. The linear LoRA on the left side is specialized to generate text tokens and the Convolutional LoRA on the right side is specialized to generate image patches. On the right handside, we show the details of convolutional operation applied to autoregressively generate image tokens. Best viewed in color.

*utilizes two distinct LoRA architectures—linear and convolutional—for text and image generation within autoregressive VLGs.*

## 3 BACKGROUND: AUTOREGRESSIVE VISION-LANGUAGE GENERALISTS

Existing autoregressive VLGs can be broadly classified into two categories: those that represent each image as a sequence of *discrete tokens* (Yasunaga et al., 2023; Aghajanyan et al., 2022; Team, 2024), and those that represent each image as a sequence of *continuous vectors* (Sun et al., 2023b; 2024). However, despite these differences in image representation, their underlying model architectures and formulations for vision-language generation remain largely similar. Thus, we do not differentiate them in the following formulation.

**Model Architecture** Autoregressive VLGs typically comprise three components: an image encoder (e.g., CLIP (Sun et al., 2023a) or VQ-VAE encoder (Gafni et al., 2022)), a decoder-only large language model (LLM), and an image decoder (e.g., a diffusion model (Podell et al., 2023) or VQ-VAE decoder (Gafni et al., 2022)). Given a sequence of interleaved text segments and images, the image encoder processes each image into a sequence of image tokens. These image tokens are then concatenated with the text tokens in their original order and input into the LLM. The LLM autoregressively predicts the next token, which could be either text or image. Finally, the image decoder takes in the predicted image tokens and reconstructs the target image.

**Training Objective** The training objective of VLGs can be loosely defined in the following unified autoregressive manner.

$$\arg\max_{\theta} \sum^{\mathcal{D}} \sum_{n=1}^{N} P_{\theta}(s_n | s_1, s_2, ..., s_{n-1}) \tag{1}$$

where $\theta$ denotes the model parameters, $N$ denotes the input sequence length, $\mathcal{D}$ denotes the training dataset, and $s_i$ denotes a text token or an image-patch embedding. This unified objective is optimized through two types of losses: (1) If the image is represented as discrete tokens, the CrossEntropy loss is employed to minimize the divergence between the predicted probability distribution of the image or text tokens and the ground truth distribution; (2) If the image is encoded as continuous vectors, the mean-squared-error (MSE) loss is used to minimize the difference between the predicted and actual image embeddings.

# 4 MODALITY-SPECIALIZED SYNERGIZERS (MoSS)

In this section, we first detail the two modality-specialized adaptations in MoSS: Linear LoRA for text tokens and Convolutional LoRA for image tokens. We then describe the process of synergistically integrating these adaptations into autoregressive VLGs to perform interleaved generation.

## 4.1 LINEAR LOW-RANK ADAPTATION (LoRA)

LoRA (Hu et al., 2021) is a parameter-efficient finetuning method that freezes the pretrained model parameters and injects low-rank decomposable matrices into the layers of transformers. Formally, given the weights in a linear layer $\mathbf{W} \in \mathbb{R}^{d_{out} \times d_{in}}$, LoRA modifies the weights by adding a decomposable weight matrix $\Delta \mathbf{W}$ to $\mathbf{W}$. Thus, for a vector $\mathbf{h} \in \mathbb{R}^{d_{in}}$, the modified linear transformation $T : \mathbb{R}^d_{in} \to \mathbb{R}^d_{out}$ becomes:

$$T(\mathbf{h}) = \mathbf{h}(\mathbf{W} + \Delta \mathbf{W})^\top = \mathbf{h}\mathbf{W}^\top + \mathbf{h}\Delta \mathbf{W}^\top \tag{2}$$

$\Delta \mathbf{W}$ is decomposed into two low-rank matrices, i.e., LoRA $A$: $\mathbf{W}_A \in \mathbb{R}^{r \times d_{in}}$ and LoRA $B$: $\mathbf{W}_B \in \mathbb{R}^{d_{out} \times r}$ satisfying the low-rank constraint $r \ll min(d_{out}, d_{in})$. The final expression is

$$T(\mathbf{h}) = \mathbf{h}\mathbf{W}^\top + \alpha \mathbf{h}\mathbf{W}_A^\top \mathbf{W}_B^\top \tag{3}$$

where $\alpha \in \mathbb{R}$ is a hyper-parameter.

## 4.2 CONVOLUTIONAL LOW-RANK ADAPTATION (CONVOLUTIONAL LoRA)

We propose Convolutional LoRA, a variant of LoRA specifically designed for modeling the local structure of image hidden states during image generation, by improving the architecture proposed in Zhong et al. (2024). Detailed empirical comparison between two Convolutional LoRAs can be found in Table 10 in Appendix C.3. The previous approach first reduces the dimension of input features and then performs the convolution operation within a lower-dimension space. Since dimension reduction can cause information loss, the convolution within a reduced dimension can be less effective at modeling the local priors of image patches. On the contrary, our method performs convolution in the original input feature space and the dimension is deducted during the convolution process, which alleviates the information loss issue in the previous design.

Specifically, our approach consists of a convolutional LoRA $A$ layer, i.e., $\text{Conv}_{k \times k}$, where the kernel size is $k \times k$, the number of input channels is $c_{in}$, and the number of output channels is $r$, as well as a LoRA $B$: $\mathbf{W}_B \in \mathbb{R}^{C_{out} \times r}$. Given the 2D feature $\mathbf{I} \in \mathbb{R}^{H \times W \times C_{in}}$ of an image, where $H$ denotes the height, $W$ denotes the width, and $C_{in}$ denotes the number of channels of $\mathbf{I}$, the convolutional LoRA $A$ projects down its number of channels to $r$ and simultaneously performs convolution operation. Then the LoRA $B$ projects its number of channels up to $C_{out}$. The equation 3 becomes:

$$\tilde{T}(\mathbf{I}) = \mathbf{I}\mathbf{W}^\top + \alpha \text{Conv}_{k \times k}(\mathbf{I})\mathbf{W}_B^\top \tag{4}$$

where $\alpha$ is a hyper-parameter.

## 4.3 INTEGRATING MoSS INTO VLGs

As shown in Figure 2, we propose to integrate two types of adaptations into VLGs, i.e., using **Linear LoRA** for text generation and **Convolutional LoRA** for image generation. Formally, let $\mathbf{W} \in \mathbb{R}^{d_{out} \times d_{in}}$ be the weights of any linear layer in a LLM, and let $\mathbf{H} = [\mathbf{h}_1^t, ..., \mathbf{h}_m^t, \mathbf{h}_{m+1}^i, \mathbf{h}_{m+2}^i, ..., \mathbf{h}_{m+(H \times W)}^i, ..., \mathbf{h}_N^t] \in \mathbb{R}^{N \times d_{in}}$ denotes the hidden states of a sequence of interleaved text and images, where a subscript indicates position of a hidden state and the superscript indicate if a hidden state is decoded into a text token ($t$) or decoded into an image-patch embedding ($i$). We untie $\mathbf{H}$ into text hidden states $\mathbf{H}^t = [\mathbf{h}_1^t, \mathbf{h}_2^t, ..., \mathbf{h}_m^t, \mathbf{h}_{m+(H \times W)+1}^t, ..., \mathbf{h}_N^t]$ and image hidden states $\mathbf{H}^i = [[\mathbf{h}_{m+1}^i, ..., \mathbf{h}_{m+(H \times W)}^i], [\mathbf{h}_{n+1}^i, ..., \mathbf{h}_{n+(H \times W)}^i], ...]$, where $m + 1$ and $n + 1$ denote the starting positions of two subsequences of image hidden states. Each subsequence of a single image has a fixed length of $H \times W$ and we reshape the hidden states of each image in $\mathbf{H}^i$ into a 2D structure. Hence, the dimension of $\mathbf{H}^i$ becomes $B \times H \times W \times C_{in}$, where $B$ denotes

the number of images in the sequence $\mathbf{H}$. We feed $\mathbf{H}^t$ into the Equation 3 to get $\hat{\mathbf{H}}^t = T(\mathbf{H}^t)$ and $\mathbf{H}^i$ into Equation 4 to get $\hat{\mathbf{H}}^i = \tilde{T}(\mathbf{H}^i)$.

It is non-trivial to integrate convolutional operation in auto-regressive model and to the best of our knowledge, we are the first to incorporate the convolutional architecture to improve interleaved generation. The right part of Figure 2 visualizes the convolutional operation applied to a sequence of image patches. The squares on the left denote the reshaped 2-dimensional input image patches and the larger blue squares denote the $2 \times 2$ convolution kernels. The number on each square denotes the original positions of a patch in the image sequence. For demonstration purposes, we draw image patches with $H = 3$ and $W = 3$. Note that the current hidden state of an image patch can only depend on previous hidden states since we use the autoregressive architecture. Thus, when applying the convolution operation on an image patch, the kernel only covers neighboring patches on the top and left sides of a patch. For example, the new hidden state of patch 9 is computed from patches: $5, 6, 8$, and 9. To preserve the shape ($H \times W$) of the input image patches, we pad the reshaped image hidden states with zero vectors on the top and left sides, as shown by the grey squares in Figure 2. Finally, we assemble $\hat{\mathbf{H}}^i$ and $\hat{\mathbf{H}}^t$ back to their original sequence to form $\hat{\mathbf{H}}$.

## 5 INTERLEAVED INSTRUCTION TUNING WITH LEAFINSTRUCT

Existing interleaved vision-language models (Sun et al., 2023b; 2024; Dong et al., 2024) predominantly follow the training procedures that they are first pretrained on massive corpora of interleaved data such as MMC4 (Zhu et al., 2023c) and other resources and then finetuned on a mix of high-quality datasets, such as visual instruction tuning data in Liu et al. (2023c) and Instruct-Pix2Pix (Brooks et al., 2023b). However, one significant limitation of these instruction-tuning datasets is that the outputs are typically in a single modality, e.g., either text or image, which hinders the instruction-following capability of VLGs especially in generating interleaved text and images specified by the given instructions.

### 5.1 DATASET: LEAFINSTRUCT

To bridge the gap between limited existing resources and the practical need for improving interleaved generation models, we curated LEAFINSTRUCT, the first comprehensive instruction tuning dataset for interleaved text-and-image generation. Each instance in our dataset consists of (1) a detailed instruction, (2) an input context with interleaved text and images, and (3) a ground-truth output also with interleaved text and images. We show an example of LeafInstruct in Figure 6, and compare it with other representative instruction-based datasets.

**Dataset construction** We construct a diverse instruction-tuning data collection from large-scale web resources and academic datasets, including MMDialog (Feng et al., 2023), VIST (Huang et al., 2016), WikiWeb2M (Burns et al., 2023) and YouCook2 (Zhou et al., 2018). Since the original data sources can be noisy, we meticulously devised an automatic data annotation pipeline to ensure the high quality of our curated data. We include the details on dataset construction in Appendix B. We also conducted a rigorous human assessment of our dataset (see Section 7).

**Dataset Statistics** After applying our rigorous data processing pipeline, we totally obtain 184,982 high-quality instances out of more than 7 million source samples. Our dataset covers a wide range of realistic instruction-tuning tasks, including multimodal document completion, multimodal dialogue, visual storytelling, multimodal script generation, and knowledge-intensive generation. In Appendix B, we show the domain distribution of LEAFINSTRUCT in Figure 7 and compare our dataset with existing datasets in Table 4. These analyses effectively demonstrate the diversity and the novelty of our dataset.

### 5.2 INSTRUCTION TUNING FOR INTERLEAVED GENERATION

With our curated LEAFINSTRUCT, we enable large-scale interleaved instruction tuning so that the model can learn how to follow human instructions to generate desired interleaved text and images. To preserve the VLG's capability obtained from pre-training, we only fine-tuned the modality-specialized adaptation layers, and the remaining parameters in the VLGs are kept frozen.

Specifically, as shown on the right side of Figure 6, given the task instruction and the interleaved context as inputs, the model is trained to autoregressively generate interleaved text tokens and images with two alternative generation modes for text and images, respectively. We use a special token  to indicate where an image occurs in the interleaved sequence. The training process is as follows: **(1)** The model is set to the **text generation** mode by default. During this mode, the hidden states of newly generated tokens are always routed to linear LoRA, and only the parameters in the linear LoRA are optimized. **(2)** After the  token is generated, the model switches to **image generation** mode. The VLG takes in the updated context ended with  and is trained to generate a fixed-length ($H \times W$) sequence of image patch embeddings autoregressively. All the hidden states of generated image embeddings are routed to Convolutional LoRA and only the parameters in the Convolutional LoRA are fine-tuned. **(3)** When the generation of an image is finished, the model is trained to predict an end-of-image token </IMG>, and the model will resume the text generation mode. This process will be iterated until the training on a sequence is finished.

**Interleaved Inference** The inference procedure of our framework is largely identical to the instruction tuning, where two generation processes iterate alternatively. The only key difference is that the fine-tuned VLGs will automatically determine when to generate a text segment or an image at their own discretion. The iterative generation process terminates when the model produces the end-of-generation token  at the end of a response. Note that although our inference process is designed for interleaved generation, we can also handle the cases where the outputs only contain text or images, enabling a wide range of applications.

## 6 EXPERIMENTS

### 6.1 EXPERIMENT SETUP

**Evaluation Benchmarks** We evaluate the interleaved generation capability of our method on InterleavedBench (Liu et al., 2024b). InterleavedBench is a comprehensive dataset specifically tailored for interleaved evaluation. InterleavedBench covers a diverse array of tasks, where the evaluation data are either curated by the authors (e.g., *document completion*), or re-annotated based on subsets of well-established academic evaluation benchmarks, including *visual storytelling* from VIST (Huang et al., 2016), *activity generation* from ActivityNet (Krishna et al., 2017), *script generation* from WikiHow (Yang et al., 2021), *image editing* from MagicBrush (Zhang et al., 2023a), and *multi-concept image composition* from CustomDiffusion (Kumari et al., 2023). We include more details on evaluation benchmarks in Appendix C.1.

**Evaluation Metrics** We adopt InterleavedEval (Liu et al., 2024b), a strong reference-free evaluation metric to conduct a holistic assessment of the quality of interleaved generation. InterleavedEval prompts GPT-4o to score an interleaved output from five aspects, including Text Quality, Perceptual Quality, Image Coherence, Text-Image Coherence (TIC), and Helpfulness. For each aspect, the GPT-4o outputs a discrete score from $\{0, 1, 2, 3, 4, 5\}$, where 0 is the worst and 5 is the best. We refer to the original paper (Liu et al., 2024b) for a detailed definition of each score and each evaluation aspect. We also have an additional evaluation on image editing on the *full test set* of MagicBrush using well-established metrics, including CLIPScore (Hessel et al., 2021) and DINO (Caron et al., 2021) in Table 5 in Appendix C.2.

**Implementation Details** To demonstrate the generalizability of our method, we adopt our MOSS to two representative autoregressive VLG backbones, i.e., Emu2 (Sun et al., 2024) and Chameleon (Team, 2024), and fine-tune them on our LEAFINSTRUCT dataset. The rank number of all the LoRA is set to 256 by default. Note that for the Chameleon model, we adopt the implementation in Chern et al. (2024) since the original model and checkpoints are not publicly available. More implementation details including hyperparameters and GPU setups can be found in Appendix A.1.

**Baselines** For fair comparisons, we primarily compare our methods with current state-of-the-art **open-source** VLGs, including GILL (Koh et al., 2023), MiniGPT-5 (Zheng et al., 2023a), Pretrained Emu2, and Chameleon. We also report the performance of pipelines based on **proprietary** models, including Gemini 1.5 (Reid et al., 2024)+SDXL (Podell et al., 2023) and GPT-4o (OpenAI, 2024)+DALLE 3 (Betker et al.). For these baselines, we first prompt the VLMs (e.g., GPT-4o) to generate text along with image captions, and then feed the image captions to a separate image generation model (e.g., DALLE). We report these performances only for reference purposes.

Table 1: **Main results of interleaved generation on InterleavedBench**. We show the performance of pipelines based on proprietary models (Top), open-source VLGs (Middle), and the VLGs trained with our MoSS and LEAFINSTRUCT (Bottom), respectively. Note that the scale is from 0 to 5 (5 is the best). We also report the percentage of improvement in our method over the original VLG backbone in the parentheses. The best results are highlighted in **bold**.

| Model | Text Quality | Perceptual Quality | Image Coherence | TIC | Helpfulness |
|---|---|---|---|---|---|
| **Proprietary Models** | | | | | |
| Gemini1.5 + SDXL | 3.37 | 4.34 | 3.34 | 3.98 | 3.28 |
| GPT-4o + DALL·E 3 | 3.16 | 4.44 | 3.13 | 4.39 | 3.46 |
| **Open-Source Models** | | | | | |
| MiniGPT-5 | 1.31 | 3.44 | 2.06 | 2.66 | 1.76 |
| GILL | 1.44 | **4.02** | 2.12 | 2.69 | 1.53 |
| Emu2 | 1.33 | 2.29 | 1.71 | 1.22 | 1.87 |
| Chameleon | **3.33** | 0.67 | 0.28 | 0.47 | 1.43 |
| Emu2 + MoSS (Ours) | 2.61 (+96.2%) | 3.62 (+58.1%) | **3.41** (+99.4%) | **3.54** (+190.2%) | **2.71** (+44.9%) |
| Chameleon + MoSS (Ours) | 2.98 (-10.5%) | 2.25 (+235.8%) | 1.05 (+275%) | 1.7 (+261.7%) | 1.82 (+27.3%) |

## 6.2 MAIN RESULTS

**Quantitative Results** Table 1 presents the main results of our method in comparison to the baselines. We have the following findings. **Firstly**, our approach is highly effective and efficient when it is adapted to existing VLGs. Applying our MoSS to VLGs achieved significant improvement over their original performance on all evaluation aspects. For example, compared with the original Emu2 model, Emu2+MoSS achieved a performance gain of **up to 190.2%** (on Text-Image Coherence) and **97.76%** on the average of 5 aspects, almost doubling the overall performance. **Secondly**, our method beats the previous open-sourced state-of-the-art (i.e., GILL) by a large margin, i.e., **34.7%** on the average of 5 aspects. Particularly, the outputs of our method have better coherence across images (w/ 37.8% improvement in Image Coherence) and between text and images (w/ 31.6% improvement in Text-Image Coherence). Our method also exhibits better instruction-following capability and is able to generate more helpful content given the 11.5% improvement in Helpfulness. **Thirdly**, it is worth noting that the Chameleon baseline achieves good performance on Text Quality but extremely poor performance on image-related aspects. We observed that Chameleon usually generates long and comprehensive text responses with no image output, thus leading to poor performance on image-related aspects. We hypothesize the reason lies in the lack of instruction tuning on interleaved generation with both text and images. From Table 1, our approach improves the original Chameleon by a significant margin, especially on image-related aspects. This shows that our interleaved instruction tuning can effectively enhance a VLG that was previously poor at mixed-modal generation. We noticed that adding MoSS in Chameleon can cause a slight performance drop in text quality. We discuss the details of this problem in Appendix C.3, conduct an additional human evaluation of text quality, and present the results and findings in Table 11. **Fourthly**, there remains a notable gap between open-sourced VLGs and the pipeline approaches based on proprietary models, indicating building a powerful and general-purpose open-sourced VLGs is still challenging.

**Per-task Performance** We also show the average performance on each task on InterleavedBench in Figure 3. Specifically, our method (i.e., Emu2-MoSS) outperforms the baselines on most tasks, often by a large margin. For subject-driven generation, the slightly lower performance of our approach compared to other baselines is due to its poorer perceptual quality. We included more detailed justifications for this result in the following section. We report the per-task performance on all aspects in Figure 9 in Appendix C.5. In addition, we include more results on well-established benchmarks of image-understanding, text-to-image generation, and image-editing in Table 5, 6, 7, 8, and 9 in Appendix C.2.

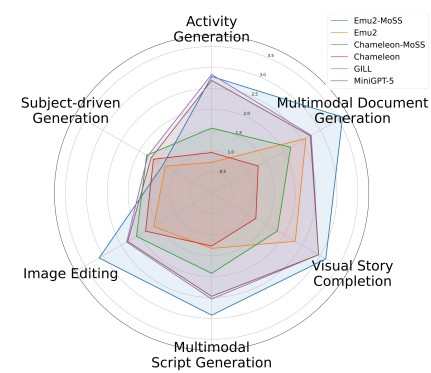

Figure 3: **Per-task performance** averaged on 5 aspects on InterleavedBench.

**Qualitative Results** To better interpret the results, we conducted a qualitative analysis on several open-sourced baselines and our MoSS in Figure 4. Our findings are as follows. **Firstly**, our method

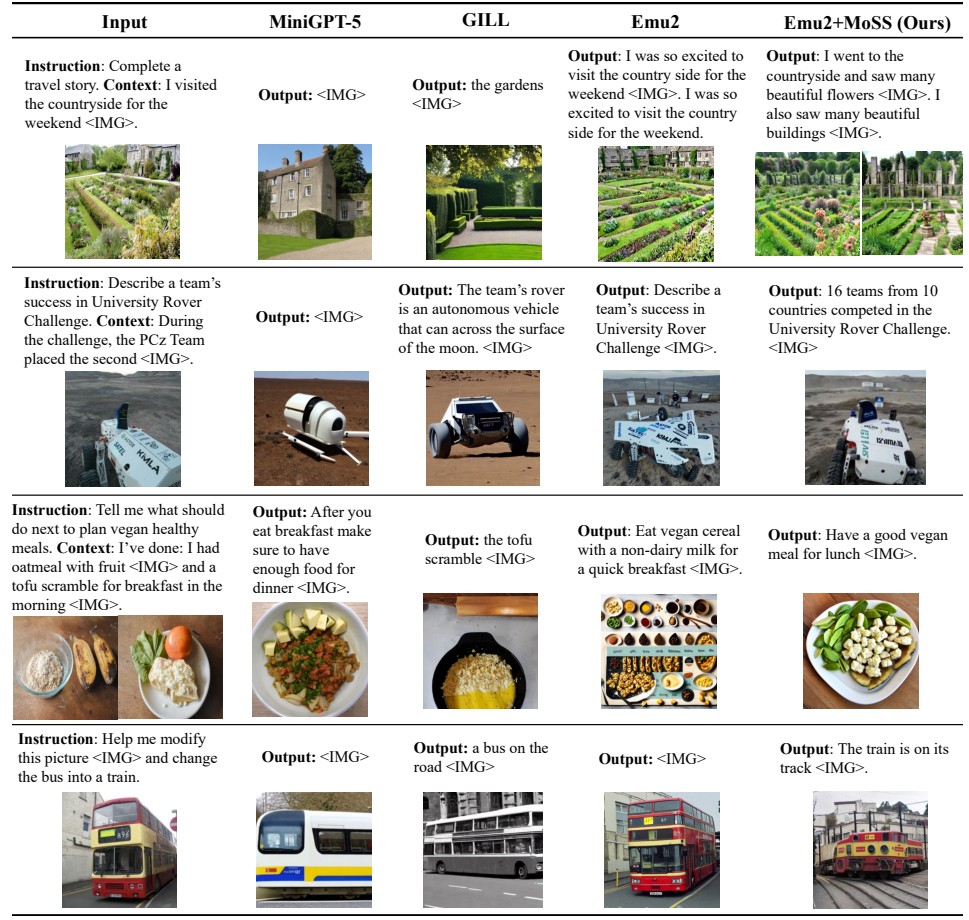

Figure 4: **Qualitative results** of MOSS based on Emu2 and open-source baselines. The  tokens denote the images' positions in the interleaved sequences.

demonstrates better helpfulness and instruction-following capabilities. For example, in the first row in Figure 4, our method generates a more coherent visual story with more diverse content given the input. In the third row, our approach provides a more natural and reasonable next step for the user, i.e., *have a good vegan lunch*, while other baselines either jump to *dinner* (MiniGPT-5) or stick to *breakfast* (Emu2). **Secondly**, we observed that poor text quality is a common issue for many baselines. For instance, MiniGPT-5 often fails to generate explanatory text while GILL usually generates a short caption, e.g., *the gardens* in the first row and *the tofu scramble* in the third row, instead of generating useful content to solve the task. **Thirdly**, neither GILL nor MiniGPT-5 can preserve the visual appearance of the entities and scenes in the input images. Our approach, on the contrary, faithfully retains most visual characteristics, leading to significantly better Image Coherence. **Finally**, as shown in the fourth row in Figure 4, for tasks such as image editing or image composition, although MiniGPT-5 and GILL can sometimes generate images with better perceptual quality, the image contents are often irrelevant to the task, ignoring input instructions and context. In contrast, our method strives to adhere to instructions and can better condition its generation on the provided image. Due to the complexity of the task, our model may produce images with lower perceptual quality and noticeable distortions. However, when taking Helpfulness into account, the images generated by our model can be considered as the better ones compared with the baselines. We present additional qualitative results of Chameleon + MOSS in Figure 8 in Appendix C.4.

## 7 DISCUSSIONS

**Comparison between MOSS and other PEFT Methods**  To directly validate the performance improvement brought by our proposed MOSS, we fine-tuned Emu2 using (1) traditional linear LoRA (Hu et al., 2021) and (2) Mixture-of-Expert (MoE) LoRA (Shen et al., 2024), with the results presented in Table 2. In traditional linear LoRA, text and images share the same low-rank

Table 2: **Comparison between MoSS and existing PEFT methods**, i.e., traditional linear LoRA, and Mixture-of-Expert (MoE) LoRA. Mixture-of-Expert LoRA uses two different sets of linear LoRA for images and text, respectively. The rank number is set to 256 for all methods in this table.

| Model | Text Quality | Image Quality | Image Coherence | TIC | **Helpfulness** |
|---|---|---|---|---|---|
| Emu2 | 1.33 | 2.29 | 1.71 | 1.22 | 1.87 |
| + LoRA | 1.77 | 2.38 | 1.99 | 2.04 | 1.64 |
| + MoE-LoRA | 1.98 | 3.28 | 2.66 | 2.62 | 2.01 |
| + MoSS (Ours) | **2.61** | **3.62** | **3.41** | **3.54** | **2.71** |

adaptation parameters, while in MoE-LoRA, two different sets of linear LoRA are used for images and text respectively. The routing strategy in MoE-LoRA is based on the output modality of each hidden state, i.e., whether the hidden state is used to generate text or image. From Table 2, we effectively verify the benefits of using separate parameters for image and text. MoSS significantly outperforms the MoE-LoRA across all aspects, especially the image-related aspects such as Image Coherence and Text-Image Coherence (TIC). The conclusions from the results are two-fold. First, it shows that introducing modality-specialized architecture and parameters can effectively improve interleaved text-and-image generation. Second, it verifies that convolutional LoRA can improve image generation by better modeling the local priors of images. Additionally, we compare the computational cost of using MoSS and LoRA in Appendix C.3.

**Effect of Rank Number** To investigate how the number of rank $r$ can affect the performance, we show the performance averaged on 5 aspects on InterleavedBench with the rank number equals $(32, 64, 128, 256)$ comparing LoRA, MoE-LoRA, and our MoSS in Figure 5. Our approach consistently outperforms LoRA and MoE-LoRA across all rank numbers, and as the rank number increases, the gap between MoSS and previous methods consistently grows larger. This proves the effectiveness and generalizability of MoSS across different rank sizes. Based on this experiment, we set the rank number of our approach to 256 by default. We include more results on the effect of rank in LoRA in Table 13, 14, and 15 in Appendix C.6.

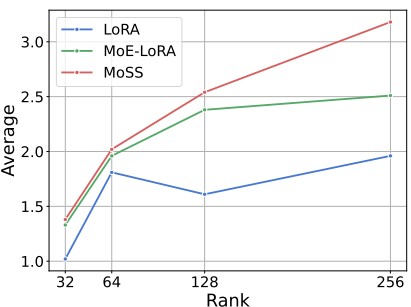

Figure 5: Performance averaged on 5 aspects with different rank numbers.

**Quality Assessment of LEAFINSTRUCT** To verify our LEAFINSTRUCT dataset is of high quality, we conduct a rigorous human evaluation using the multi-aspect evaluation criteria in InterleavedEval (Liu et al., 2024b). Specifically, we use a scale of 0 to 3 in the evaluation, where 0 is the lowest score while 3 is the highest. We randomly sampled 200 instances from LEAFINSTRUCT and asked two human annotators with expertise in NLP and multimodal research to rate each instance from 5 aspects. We report the averaged scores from two annotators in Table 3. We show that the sampled instances consistently achieved almost full scores across all 5 aspects, which effectively demonstrated that our curated dataset is of high quality.

Table 3: **Human evalution** of randomly sampled instances from LeafInstruct. Note that the scale is from 0 to 3 (**Score 3 is the best**), which is different from the scale used in Table 1 and Table 2.

| | Text Quality | Perceptual Quality | Image Coherence | TIC | Helpfulness |
|---|---|---|---|---|---|
| Score | 2.89 | 2.96 | 2.77 | 2.87 | 2.71 |

## 8 CONCLUSION

We propose MODALITY-SPECIALIZED SYNERGIZERS (MoSS), a novel modality-specialized adaptation framework tailored for VLGs. MoSS dedicates a set of linear LoRA for processing text and a set of Convolutional LoRA for images, allowing each modality to have its own optimal adaptation design. Besides, we propose the first interleaved instruction tuning dataset LEAFINSTRUCT and verify the dataset quality via rigorous human evaluation. Extensive experiments on InterleavedBench showcase that our proposed method and dataset are highly effective, establishing the new state-of-the-art among open-sourced VLGs in interleaved text-and-image generation.

## ACKNOWLEDGEMENT

This research is partially supported by a research award from Intuit AI Research and the award No. #2238940 from the Faculty Early Career Development Program (CAREER) of the National Science Foundation (NSF). The views and conclusions contained herein are those of the authors and should not be interpreted as necessarily representing the official policies, either expressed or implied, of the U.S. Government. The U.S. Government is authorized to reproduce and distribute reprints for governmental purposes notwithstanding any copyright annotation therein.

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

## A  MORE DETAILS OF MOSS

### A.1  IMPLEMENTATION DETAILS

We leverage the Emu2 model (Sun et al., 2024), consisting of the EVA-02-CLIP-E-plus (Sun et al., 2023a) as the image encoder, the LLaMA-33B (Touvron et al., 2023), and the SDXL (Podell et al., 2023) as the image decoder, as our base model. The EVA-02-CLIP-E-plus and the LLaMA-33B is connected by a linear project-up layer and the LLaMA-33B and the SDXL is connected by a linear project-down layer. All the variants of LoRA in Section 7, including our MoSS are trained with LEAFINSTRUCT for one epoch on $8 \times$ A100 GPUs with learning rate $2e^{-5}$, batch size 1 per GPU, and a gradient accumulation step of 16. All the LoRA have a rank of 256, dropout rate of $0.05$, and the LoRA $\alpha$ in Section 4 is set to $2 \times 128$. The kernel size of MoSS is $2 \times 2$, the stride is set to 1. During training, all parameters of the Emu2 model are kept frozen and only the LoRA parameters are updated.

## B  MORE DETAILS OF LEAFINSTRUCT

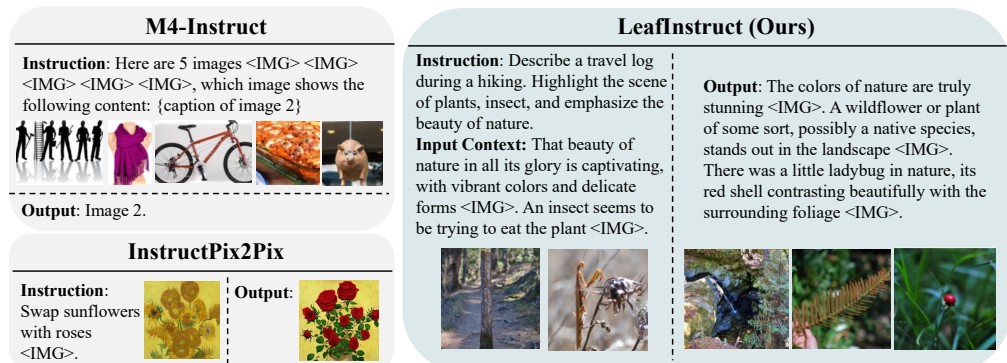

Figure 6: Comparison between existing benchmarks and our LEAFINSTRUCT. In existing datasets such as InstructPix2Pix (Brooks et al., 2023b) and Mantis-Instruct (Li et al., 2024a), the outputs are in single modality, either text or image. On the contrary, the inputs and outputs of our LEAFINSTRUCT cover multiple modalities.

Table 4: Comparison between our LEAFINSTRUCT and existing instruction tuning datasets.

| Dataset Name | Input Text | Input Images | Output Text | Output Images | Publicly Available |
|---|---|---|---|---|---|
| LLaVA (Liu et al., 2023c) | Yes | Single | Single | No | Yes |
| MultiInstruct (Xu et al., 2023) | Yes | Single | Single | No | Yes |
| Vision-Flan (Xu et al., 2024) | Yes | Single | Single | No | Yes |
| InstructPix2Pix (Brooks et al., 2023a) | Yes | Single | No | Single | Yes |
| MagicBrush (Zhang et al., 2023a) | Yes | Single | No | Single | Yes |
| SuTI (Chen et al., 2023c) | Yes | Multiple | No | Single | No |
| Instruct-Imagen (Hu et al., 2024) | Yes | Multiple | No | Single | No |
| Mantis-Instruct (Jiang et al., 2024) | Yes | Multiple | Yes | No | Yes |
| LEAFINSTRUCT (Ours) | Yes | Multiple | Yes | Multiple | Yes |

### B.1  MORE DETAILS IN DATASET CONSTRUCTION

We elaborate on the details of our dataset construction pipeline as follows. **Firstly**, we filter the samples based on the text length, number of images, and the coherence between text and images (measured by CLIPScore (Hessel et al., 2021)). We only keep the instances with 3 to 6 images in total. We also discard the instances with more than 12 sentences to ensure a balanced ratio between the number of textual sentences and images. **Secondly**, we leverage a state-of-the-art open-sourced LLM (i.e., `Llama-8B-Instruct`) as a text filter to discard the instances with poor text quality. **Thirdly**, we remove the instances with duplicate or perceptually highly similar images to ensure

the diversity of the images. **Finally**, we also apply Llama3 to annotate the task instruction for each instance based on the text content and rewrite the text if it's too verbose to prevent the context length from being too long.

**Details of Text Quality Filter** We use `Llama-8B-Instruct` model to rate the text quality of an instance with the following prompt: *"Imagine you are an expert data annotator. You are given a text material and you need to evaluate its quality in terms of whether it is coherent, fluent, easy to understand, and helpful to humans. Please be critical and rate the quality as good only when the text quality is good in all four aspects. Output 1 if you think the material is good after you consider all four aspects. Output 0 if you think the material is not good enough. Here is the text material to be evaluated: {TEXT} Only output 0 or 1 and do not output anything else. Your evaluation is:"* We discard the instances if the output from Llama is 0.

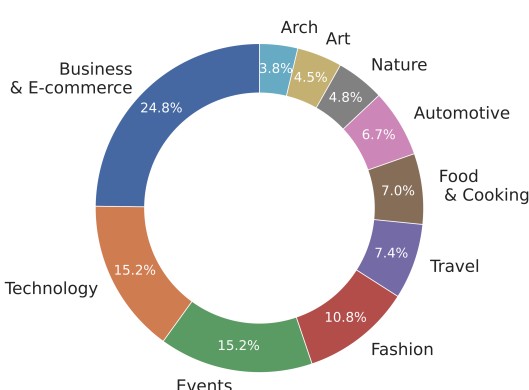

Figure 7: Domain distribution in LeafInstruct.

**Details of Image Filter** We empirically found that if the images are too identical in the training instances, the trained models tend to find a shortcut to simply copy the image during generation. To this end, we design a filter to discard the instances with duplicate images to improve data quality. Specifically, we leverage the LPIPS score (Zhang et al., 2018) that measures the perceptual similarity between the images. Specifically, for each instance, we enumerate each pair of images and compute their LPIPS score. If there is one pair with a score higher than 0.6, we discard the instance. We determine the threshold of 0.6 by empirical trial.

**Details of Instruction Annotation** We also adopt `Llama-8B-Instruct` to annotate the task instruction for each instance. We devise instructions to prompt the Llama3 model to rewrite the original text material in the pretraining dataset MMC4 into instruction-tuning instances. The input context length is 2048 and the output context length is 1024. We set the temperature as 1 to encourage the diversity of instructions. We use the following prompt: *"Imagine you are an expert instruction annotator. You are given a material. You need to read its content and output a brief task instruction with one sentence such that another person can recover the given the material given the instruction. The instruction you predict should be specifically tailored for creative interleaved content generation that consists of both text and images. Now you need to annotate a concise, accurate instruction for the following instance. Please only predict the instruction and do not output anything else. Please design the instruction for the multi-modal generation task interleaved with both text and images. Text: {TEXT} Instruction:"*.

## C   MORE EXPERIMENT RESULTS

### C.1   MORE DETAILS ON EVALUATION BENCHMARKS

InterleavedBench has two splits: a context-based split in which the input of each instance is equipped with interleaved text and images; and a context-free split with text-only inputs. The context-based split contains 465 instances and the text-only split contains 350 instances. We only use the context-based split as the testing set since we mainly focus on tasks with interleaved inputs and outputs.

### C.2   MORE RESULTS ON ESTABLISHED BENCHMARKS

**Interleaved Generation and Image Editing** Although InterleavedBench is a new evaluation benchmark, it also consists of testing instances from 3 well-established benchmarks including (1) visual story completion from VIST, (2) MagicBrush, and (3) multi-concept image composition from CustomDiffusion. Below, we directly report the performance of MoSS and baselines on these three

well-established benchmarks in Table 6, 7, 8, respectively. We also report the performance of Mag-icBrush using established metrics in Table 5.

Table 5: **Results of image editing on the full test set of MagicBrush**. We show the performance of open-source VLGs (Top), and the VLGs trained with our proposed method (Bottom), respectively.

| Model | CLIP-I | DINO | CLIP-T | AVG |
|---|---|---|---|---|
| MiniGPT-5 | 72.04 | 41.66 | 25.45 | 46.38 |
| GILL | 72.95 | 43.32 | 24.81 | 47.03 |
| SEED-X-Edit (Ge et al., 2024) | 85.56 | 68.74 | **27.28** | 60.53 |
| Emu2 + MoSS (Ours) | **85.88** | **74.92** | 25.98 | **62.26** |

**Multimodal Understanding and Text-to-Image Generation** To show that our MoSS framework can also excel on tasks requiring single modality outputs i.e., the output only contains text or an image, we evaluate its performance on widely adopted image understanding benchmarks including MMBench, MME, MMMU, Pope, and MM-Vet, and text-to-image generation benchmarks includ-ing MSCOCO 30K (Lin et al., 2014), and GenEval (Ghosh et al., 2024). For MSCOCO-30K, following the previous evaluation protocol (Sun et al., 2024), we randomly sample 30,000 captions from the validation set of MSCOCO and generate 30,000 images. We report the FID between the 30,000 generated images and real images from the validation set of MSCOCO (Note for FID, the lower the better). For other benchmarks, we adopt their official implementation of the evaluation. Since LeafInstruct mainly targets tasks with interleaved outputs, we augmented it with 500,000 instances from Vision-Flan (Xu et al., 2024), a popular visual-instruction tuning dataset targeting image understanding, and 500,000 instances from LAION-COCO[2], a standard training dataset for text-to-image generation. We finetune Emu2 with LoRA, MoE-LoRA, and MoSS on the mixed dataset. We report their performance in Table 9.

Table 6: Performance of our MoSS, traditional linear LoRA, and Mixture-of-Expert (MoE) LoRA using Emu2 as the backbone model on the VIST subset in InterleavedBench.

| PEFT | Text Quality | Image Quality | Image Coherence | TIC | **Helpfulness** |
|---|---|---|---|---|---|
| LoRA | 0.3 | 0.52 | 0.46 | 0.68 | 0.43 |
| MoE-LoRA | **1.76** | 2.19 | 2.52 | 2.91 | 1.92 |
| MoSS | 1.73 | **2.87** | **2.54** | **3.30** | **2.26** |

Table 7: Performance of our MoSS, traditional linear LoRA, and Mixture-of-Expert (MoE) LoRA using Emu2 as the backbone model on the MagicBrush subset in InterleavedBench.

| PEFT | Text Quality | Image Quality | Image Coherence | TIC | **Helpfulness** |
|---|---|---|---|---|---|
| LoRA | N/A | 2.33 | 1.48 | N/A | 0.93 |
| MoE-LoRA | N/A | 2.85 | 1.54 | N/A | 1.08 |
| MoSS | N/A | **3.43** | **2.03** | N/A | **1.27** |

Table 8: Performance of our MoSS, traditional linear LoRA, and Mixture-of-Expert (MoE) LoRA using Emu2 as the backbone model on the CustomDiffusion subset in InterleavedBench.

| PEFT | Text Quality | Image Quality | Image Coherence | TIC | **Helpfulness** |
|---|---|---|---|---|---|
| LoRA | N/A | 2.95 | 1.73 | N/A | 1.56 |
| MoE-LoRA | N/A | **3.36** | 1.67 | N/A | 1.39 |
| MoSS | N/A | 3.24 | **2.04** | N/A | **1.83** |

---

[2]https://huggingface.co/datasets/laion/laion-coco

Table 9: Results on widely adopted multimodal understanding and text-to-image generation benchmarks. Note that the FID metric on MSCOCO is the lower the better.

| Model | MMBench | MME | MMMU | Pope | MM-Vet | MSCOCO-30K FID ($\downarrow$) | GenEval |
|---|---|---|---|---|---|---|---|
| Chameleon | 32.7 | 604.5 | **38.8** | 59.8 | 9.7 | 26.7 | **39.0** |
| Emu2+LoRA | 54.1 | 1148.0 | 33.7 | 87.3 | 31.3 | 23.4 | 26.8 |
| Emu2+MoE-LoRA | 54.6 | 1170.3 | 34.1 | 88.1 | 31.9 | 22.7 | 28.1 |
| Emu2+MoSS(Ours) | **56.0** | **1278.4** | 35.8 | **87.6** | **34.1** | **18.2** | 28.9 |

From Table 9, our MoSS outperforms previous LoRA and MoE-LoRA on most of the multimodal understanding benchmarks by a notable margin, which demonstrates that MoSS can be well generalized to diverse multimodal comprehension tasks. For text-to-image generation, our MoSS achieves better performance on both benchmarks, showing the effectiveness and generalizability of our approach. Notably, our MoSS achieves significantly better FID on MSCOCO-30K, which validates that our ConvLoRA can effectively improve the quality of generated images.

## C.3    ADDITIONAL ANALYSIS

**Comparison between previous and our ConvLoRA**    To show the benefits of our modified ConvLoRA architecture compared to the ConvLoRA proposed in Zhong et al. (2024) denoted as SAM-ConvLoRA, we replace the ConvLoRA in MoSS with SAM-ConvLoRA. Specifically, we set the rank of project-down and project-up matrices in SAM-ConvLoRA to 256 which is the same number of ranks in our proposed MoSS-ConvLoRA, and adopt the multi-scale convolution kernels to the size of 2x2 and 4x4. As shown in Table 10, our MoSS-ConvLoRA consistently outperforms the previous SAM-ConvLoRA on all other evaluation aspects, which demonstrates the superiority of our proposed ConvLoRA architecture. Particularly, our MoSS-ConvLoRA achieves notably better visual qualities, including perceptual quality and image coherence, thanks to our novel design that our convolution operation is applied to the full-rank original image features instead of low-rank image features as in SAM-ConvLoRA.

Table 10: Comparison of two types of ConvLoRA.

| Model | Text Quality | Image Quality | Image Coherence | TIC | **Helpfulness** |
|---|---|---|---|---|---|
| MoSS w/ SAM-ConvLoRA | 2.50 | 3.33 | 3.17 | 3.50 | 2.41 |
| MoSS w/ MoSS-ConvLoRA (Ours) | **2.61** | **3.62** | **3.41** | **3.54** | **2.71** |

**Human Evaluation on Text Quality of Chameleon and Chameleon + MoSS**    We noticed that adding MoSS in Chameleon can cause a slight performance drop in text quality. This is because the original Chameleon usually generates long and verbose text responses but with no image output. On the contrary, as the text responses in our LeafInstruct dataset are more concise to allow for including more images, after interleaved instruction tuning, our model learns to generate more concise text responses. Specifically, the average generated word length of the original Chameleon is 653, whereas that of Chameleon-MoSS is 166. The verbose responses from the original Chameleon are preferred by the LLM judge due to their verbosity bias (Zheng et al., 2023b; Saito et al., 2023), leading to a slight drop in the text quality of Chameleon-MoSS in LLM-based evaluation.

To better support this analysis, we further conduct a human evaluation of the text quality of the two models by randomly sampling 100 instances from InterleavedBench. We ask a human annotator to select the preferred text responses given the system outputs from two models. We report the Win-Tie-Loss results in Table 11. Win means our Chameleon-MoSS is better than the original Chameleon, Tie means the quality of two responses is equally good, and Loss means the original Chameleon is better.

From Table 11, the text quality of our Chameleon-MoSS is actually better than the original Chameleon. One issue we frequently observed in the original Chameleon is the text responses are overly verbose and sometimes even severely repetitive. In our evaluation protocol of text quality adopted from InterleavedEval (Liu et al., 2024b;a), such verbosity and repetitiveness are not penal-

Table 11: Human evaluation results on text quality of the original Chameleon and our Chameleon-MoSS. "Win" indicates our Chameleon-MoSS's responses are preferred by humans.

| Wins | Ties | Losses |
|------|------|--------|
| 28 | 54 | 18 |

ized, making the automatic evaluation results heavily biased towards the longer responses from the original Chameleon.

**Performance of full-finetuning**    We compare the performance of full-parameter fine-tuning using LEAFINSTRUCT in Table 12. The first row represents the results of fully fine-tuning Emu2 on our proposed LEAFINSTRUCT dataset. The second row shows the results of parameter-efficient fine-tuning Emu2 using our proposed MoSS framework. As observed, while full fine-tuning allows Emu2 to achieve better performance on text generation, the model demonstrates inferior performance on image generation due to its lack of inductive bias. In contrast, tuning with MoSS, which incorporates ConvLoRA, significantly improves image generation performance, even though the number of trained parameters in full fine-tuning is substantially larger than that of MoSS. These results clearly highlight the advantages of integrating ConvLoRA into the transformer architecture for processing visual information.

Table 12: Comparison between full finetuning and parameter-efficient tuning with MoSS based on Emu2.

| Model | Text Quality | Perceptual Quality | Image Coherence | TIC | Helpfulness |
|-------|--------------|--------------------|-----------------|------|-------------|
| Full Finetuning | 3.20 | 3.21 | 2.98 | 3.60 | 3.23 |
| MoSS (Ours) | 2.61 | 3.62 | 3.41 | 3.54 | 2.71 |

**Computation cost of ConvLoRA**    We compared the computational cost of using linear LoRA and our ConvLoRA, respectively. We compute the inference time for generating 1,000 images for each model. The total inference times for linear LoRA and ConvLoRA are 4,380 seconds and 5,910 seconds, respectively. The difference between the two models is around 1.5 seconds per image, indicating the computational cost increased by ConvLoRA is not significant.

## C.4    ADDITIONAL QUALITATIVE RESULTS OF CHAMELEON

We show the additional qualitative comparison between the original Chameleon and our Chameleon+MoSS in Figure 8. From the results, most of the examples of the original Chameleon only contain text without any image, which indicates the original Chameleon often fails to follow the instructions to generate desired mixed-modal content. Our Chameleon+MoSS, on the contrary, achieves much better performance as it is able to generate high-quality and coherent text and images. These results further validate the effectiveness of our approach and its strong generalizability and robustness when applying it to different VLGs, including Emu2 based on continuous image tokens and Chameleon based on discrete image tokens.

## C.5    PER-TASK PERFORMANCE ON EACH EVALUATION ASPECT

We report the performance on all aspects for each task in InterleavedBench in Figure 9.

## C.6    MORE RESULTS ON THE EFFECT OF RANKS

We conducted experiments to show the performance of two LoRA baselines and Lateralization LoRA on InterleavedBench with different ranks (64, 128, 256) in Table 13, 14, 15, respectively.

| Input | Chameleon | Chameleon+MoSS (Ours) |
|---|---|---|
| **Instruction**: Complete a travel story. **Context**: I visited the countryside for the weekend . | **Output**: I've always wanted to visit the coast and see the ocean. I've heard great things about the coast of Maine, so I would like to visit there. I would like to go hiking, see some lighthouses, and enjoy the fresh seafood… | **Output**: There are many popular activities to do in the countryside, including hiking, biking, camping, and visiting local attractions like museums and castles., including cycling the ridgeway Trail , camping in the yorkshire dales , and taking a boat tour of the Lake District . |
| **Instruction**: Describe a team's success in University Rover Challenge. **Context**: During the challenge, the PCz Team placed the second . | **Output**: The event challenges teams to design, build, and operate a rover that can navigate across a challenging Martian-like terrain, while performing tasks such as reconnaissance, exploration, and scientific research…. | **Output**: This competition encourages universities to design, build, and test rovers that can operate on the Martian surface, replicating real-world scenarios that engineers and scientists may encounter during future Mars missions.  |
| **Instruction**: Tell me what should do next to plan vegan healthy meals. **Context**: I've done: I had oatmeal with fruit  and a tofu scramble for breakfast in the morning . | **Output**: Have a healthy snack for breakfast, such as apple or an orange, or some almonds or cashews. | **Output**: Here are a few more ideas: Smoothie bowls made with fruit, nuts, and seeds . Avocado toast with scrambled tofu . Overnight oats made with rolled oats, milk, and fruit Chia seed pudding with fruit and nuts . Muffins or cakes made with whole grains and plant-based milk . |
| **Instruction**: Help me modify this picture  and change the bus into a train. | **Outputs**: Here's an edited version of the picture with the bus replaced by a train:  | **Outputs**: Of course! I'll make sure to do my best to assist you.  |

Figure 8: **Qualitative results** of MoSS based on Chameleon and the original Chameleon. The  tokens denote the images' positions in the interleaved sequences.

Table 13: Performance of our MoSS, traditional linear LoRA, and Mixture-of-Expert (MoE) LoRA using Emu2 as the backbone model with rank $r = 64$.

| PEFT | Text Quality | Image Quality | Image Coherence | TIC | **Helpfulness** |
|---|---|---|---|---|---|
| LoRA | 1.7 | 1.6 | 1.81 | 1.90 | 1.39 |
| MoE-LoRA | **1.86** | 2.17 | 2.17 | **2.15** | **1.66** |
| MoSS | 1.46 | **2.34** | **2.20** | 2.13 | 1.58 |

Table 14: Performance of our MoSS, traditional linear LoRA, and Mixture-of-Expert (MoE) LoRA using Emu2 as the backbone model with rank $r = 128$.

| PEFT | Text Quality | Image Quality | Image Coherence | TIC | **Helpfulness** |
|---|---|---|---|---|---|
| LoRA | 1.25 | 1.43 | 1.61 | 1.79 | 1.30 |
| MoE-LoRA | 1.94 | 2.22 | 2.42 | 2.54 | 1.90 |
| MoSS | **1.95** | **2.41** | **2.64** | **2.81** | **2.05** |

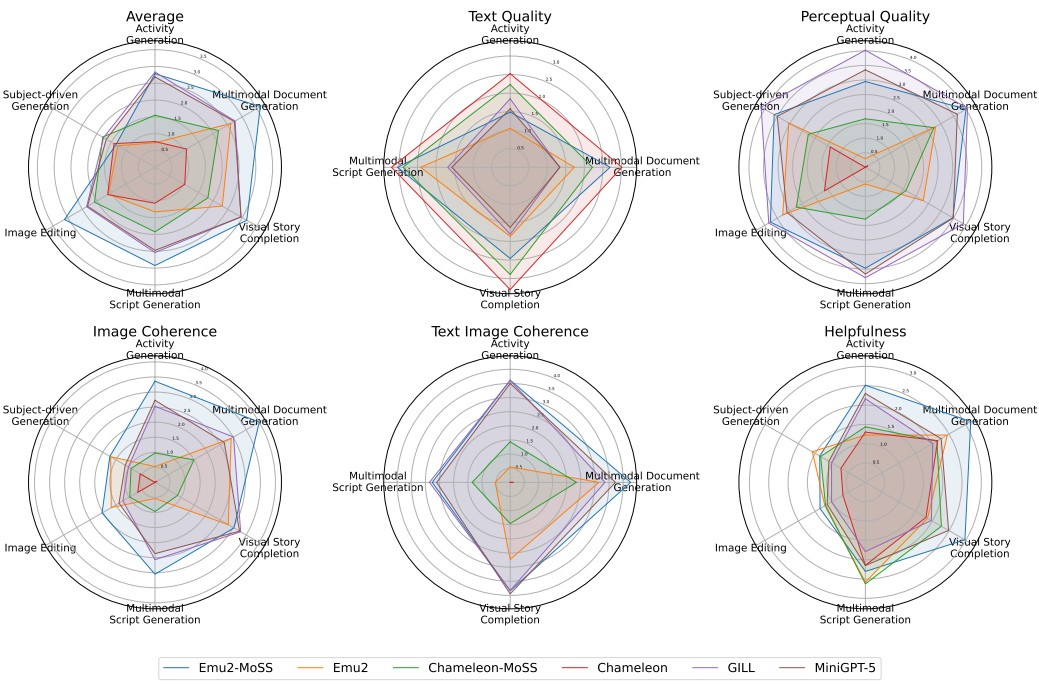

Figure 9: Per-task performance on each evaluation aspect on InterleavedBench.

Table 15: Performance of our MOSS, traditional linear LoRA, and Mixture-of-Expert (MoE) LoRA using Emu2 as the backbone model with rank $r = 256$.

| PEFT | Text Quality | Image Quality | Image Coherence | TIC | Helpfulness |
|------|--------------|---------------|-----------------|-----|-------------|
| LoRA | 1.77 | 2.38 | 1.99 | 2.04 | 1.64 |
| MoE-LoRA | 1.98 | 3.28 | 2.66 | 2.62 | 2.01 |
| MOSS | **2.61** | **3.62** | **3.41** | **3.54** | **2.71** |

