# OpenReview forum: "Modality-Specialized Synergizers for Interleaved Vision-Language Generalists"
_ICLR.cc/2025/Conference — ICLR 2025 Poster_

### Official Review · Reviewer_ccUt · 2024-10-25

**Soundness:** 3
**Presentation:** 3
**Contribution:** 3
**Rating:** 6
**Confidence:** 3

**Summary:**

This work aims to enhance the interleaved text-image generation capabilities of VLGs. The authors note that current VLGs use the same architecture for processing and generating both text and images, which may not adequately capture the distinct inductive biases inherent to each modality. To address this, they propose the Modality-Specialized Synergizers (MOSS), introducing modality-specific parameters within a unified model architecture. Specifically, they integrate convolutional LoRA for image processing and Linear LoRA for text processing.

**Strengths:**

1. This work proposes to address the distinct inductive biases of each modality by using specialized LoRA parameters.
2. The proposed method is intuitive and straightforward.
3. A new high-quality interleaved instruction tuning dataset with 184,982 instances covering over 10 domains is introduced.
4. The study conducts experiments on two different VLG backbones with both discrete and continuous image token spaces.

**Weaknesses:**

1. There are existing works focusing on interleaved image-text generation that the authors have overlooked, such as [1-3]. These works are not included in the experimental comparisons.

     [1] MM-Interleaved: Interleaved Image-Text Generation via Multi-modal Feature Synchronizer
     [2] OpenLEAF: Open-Domain Interleaved Image-Text Generation and Evaluation
     [3] Anole: An Open, Autoregressive and Native Multimodal Models for Interleaved Image-Text Generation

2. In Figure 1, the authors only show the performance of continuous embedding-based VLGs for interleaved text-image generation. What about discrete-based VLGs?

**Questions:**

1. Could the authors elaborate on why, even after fine-tuning with interleaved text-image data, a unified model is still unable to capture modality-specific inductive biases?

2. What is the relationship between the capability for interleaved image-text generation and the use of discrete tokens versus continuous embeddings? Which approach holds a distinct advantage?

---

> ### Author Response · Authors · 2024-11-23
> **Official Responses to Reviewer ccUt**
>
> We thank Reviewer ccUt for the constructive comments and valuable insights to improve this work. The responses to the comments are as follows:
>
> **W1: Comparison with more baselines**
>
> While we appreciate the reviewer for suggesting the additional baselines, we argue that we have tried our best to cover most of the mentioned baselines.
>
> (1) We did not include MM-Interleaved in the baselines because the authors only released the pre-trained checkpoint without supervised fine-tuning or instruction tuning. The code and data for supervised fine-tuning reported in their paper are not publicly available. Thus, we cannot reproduce their method.
>
> (2) OpenLEAF is a pipeline-based model where GPT-4 is cascaded with an image generation model (SDXL). It first applies GPT-4 to generate text and visual prompts and then uses SDXL to generate images using the visual prompts. This pipeline is very similar to the pipeline-based baselines (i.e., Gemini1.5+SDXL and GPT-4o+DALLE3) reported in our paper. Also, OpenLEAF is not publicly released and there are no implementation details to replicate the model.
>
> (3) Since the Chameleon checkpoint released by Meta does not include the part of the weights required for image generation, Anole introduces an efficient training method to train those missing weights, thereby enabling Chameleon's multimodal generation capability. As mentioned in the paper (Lines 361–363), we utilize the model and checkpoints provided by Anole as the implementation of Chameleon. Consequently, the reported results for Chameleon in our work are actually derived from the Anole model. We will clarify this detail in the revised version of the paper.
>
> **W2: Illustrations of discrete-based VLGs in Figure 1**
>
> Thanks for pointing this out. For VLGs based on discrete tokens, we also observed the limitations illustrated in Figure 1, including inferior text and image quality and weak instruction-following capability. We are preparing the updated version of our paper by considering all the reviewers' comments including adding the example of VLGs based on discrete tokens. We will upload the new version of the paper soon.
>
> **Q1: Why fine-tuning unified models cannot capture modality-specific inductive biases**
>
> Several recent studies [1,2] have demonstrated that the plain transformer architecture lacks vision-specific inductive biases, even after extensive fine-tuning on large-scale datasets. This is because the architecture of multi-head self-attentions is more suitable for modeling long-range global dependency and less effective at modeling local priors due to its weak inductive bias [4,5]. Through the lens of Fourier analysis [1,4,5], i.e., analyzing the amplitude of Fourier-transformed image features generated by either purely transformer-based vision models or convolution-based vision models, previous studies show that transformer-based models reduce the high-frequency signals in images, acting as a low-pass filter, and conversely, convolution-based encoders amplify high-frequency components, acting as a high-pass filter. Thus, relying solely on multi-head self-attentions can cause the VLGs to miss important local visual information,  and two structures can be combined to capture richer information from images. In our work, we harmonize two structures by integrating convolutional LoRA into the multi-head self-attention layers for effectively modeling both global and local dependency of image features in image generation.
>
> **Q2: Whether discrete tokens or continuous embeddings is better**
>
> Our work primarily focuses on enhancing pre-trained multimodal models with specialized architectures for parameter-efficient interleaved visual instruction tuning. We demonstrate that MOSS is broadly applicable to both continuous and discrete image tokens. While an in-depth discussion on the merits of discrete tokens versus continuous embeddings is beyond the scope of our work, we can provide some insights from a recent study [3], which found that continuous tokens generally outperform discrete tokens in terms of performance.
>
> **Reference**
>
> [1] Convolution Meets LoRA: Parameter Efficient Finetuning for Segment Anything Model. Zhong et al., ICLR 2024.
>
> [2] Vision Transformer Adapter for Dense Predictions. Chen et al., ICLR 2023.
>
> [3] Fluid: Scaling Autoregressive Text-to-image Generative Models with Continuous Tokens. Fan et al., 2024.
>
> [4] Inception Transformer. Si et al., NeurIPS 2022.
>
> [5] How Do Vision Transformers Work? Park et al., ICLR 2022.

---

> > ### Comment · Reviewer_ccUt · 2024-11-25
> >
> > Thanks for the insightful explanation about the Q1 and Q2.
> > I keep my score to the positive one.

---

> > > ### Author Response · Authors · 2024-11-25
> > >
> > > Thank you for your positive response and your time and effort in improving our work.

---

### Official Review · Reviewer_YfET · 2024-11-02

**Soundness:** 3
**Presentation:** 2
**Contribution:** 3
**Rating:** 6
**Confidence:** 3

**Summary:**

This paper examines the intrinsic inductive biases present in vision and language modalities in VLGs. It introduces MoSS, a novel approach that optimizes existing unified VLG architectures through modality-specialized adaptation layers—ConvLora for capturing local priors in image patches and LinearLora for handling sequential text. Additionally, the paper presents LeafInstruct, an open-source interleaved instruction tuning dataset. Experimental results demonstrate that MoSS enhances VLG model performance.

**Strengths:**

1. The paper proposes a novel idea that parameters for processing information of different modalities in VLGs should be trained with different strategies.
2. The proposed MoSS method brings promising enhancement in model performance of VLGs.

**Weaknesses:**

1. The performance improvement shown in Table 1 is inconsistent, with Chameleon displaying a decline in text quality after training with MoSS.

2. It remains unclear whether the observed performance enhancements are attributable to the MoSS training method or the LeafInstruct dataset (see Questions).

3. The parameters of ConvLora cannot be merged into the original parameters, as convolution is not linear. This limitation may lead to increased computational costs during inference.

**Questions:**

I wonder whether the results for the four middle rows in Table 1 reflect the model's original performance or its performance after full-parameter fine-tuning on LeafInstruct. Did you try full-parameter fine-tuning using your constructed data? Or full-parameter fine-tuning with two different sets of parameters for image and text tokens?

---

> ### Author Response · Authors · 2024-11-23
> **Official Responses to Reviewer YfET (Part 1)**
>
> We thank the reviewer for their constructive comments and valuable insights to improve this work. The responses to your comments are as follows:
>
> **W1: Performance drop in text quality on Chameleon-MoSS**
>
> The reason why adding MoSS in Chameleon can cause a slight performance drop in text quality is that the original Chameleon usually generates long and verbose text responses but with no image output. On the contrary, as the text responses in our LeafInstruct dataset are more concise to allow for including more images, after interleaved instruction tuning, our model learns to generate more concise text responses. Specifically, the average generated word length of the original Chameleon is 653, whereas that of Chameleon-MoSS is 166. The verbose responses from the original Chameleon are preferred by the LLM judge due to their verbosity bias [1,2], leading to a slight drop in text quality of Chameleon-MoSS in LLM-based evaluation.
>
> To better support this analysis, we further conduct a human evaluation of the text quality of the two models by randomly sampling 100 instances from InterleavedBench. We ask a human annotator to select the preferred text responses given the system outputs from two models. We report the Win-Tie-Loss results in Table A. Win means our Chameleon-MoSS is better than the original Chameleon, Tie means the quality of two responses is equally good, and Loss means the original Chameleon is better.
>
> **Table A: Human evaluation results on text quality of the original Chameleon and our Chameleon-MoSS. "Win" indicates our Chameleon-MoSS's responses are preferred by humans.**
>
> | Wins | Ties | Losses |
> |:------:|:------:|:--------:|
> |  28  |  54  |   18   |
>
> From the results, the text quality of our Chameleon-MoSS is actually better than the original Chameleon. One issue we frequently observed in the original Chameleon is the text responses are overly verbose and sometimes even severely repetitive. In our evaluation protocol of text quality adopted from InterleavedEval [3], such verbosity and repetitiveness are not penalized, making the automatic evaluation results heavily biased towards the longer responses from the original Chameleon. In addition, we also provide a qualitative comparison between the original Chameleon and Chameleon-MOSS, which also shows that applying MoSS will not hurt text quality and can alleviate the repetition issue.
>
> *Example:*
>
> *Instruction:* Describe the new features of Trek's expanded Slash lineup, highlighting the availability of carbon and aluminum models and the introduction of a more affordable top-of-the-line option.
> Input Context: Trek expands its award-winning Slash lineup in Waterloo, WI.
>
> *Original Chameleon Response:* ``The new features include the introduction of the Alpha Platinum Aluminum model, a lower-priced carbon version, and upgrades across all models. The current price range for the new models is $5,000 to $20,000. The features of the new full-carbon Slash 9.9 mountain bike include carbon wheels and Rapid Drive hub. The price range for the new full-carbon Slash 9.9 mountain bike is $20,000 to $25,000. The price range for the new aluminum Slash 9.9 mountain bike is $10,000 to $15,000……``
>
> *Chameleon-MoSS Response (ours):* ``There are several options in Trek's Slash lineup. For example, you can now purchase the new top-of-the-line Slash 9.9 Alpha Platinum Aluminum, which comes equipped with top-notch components such as an SRAM X01 Eagle drivetrain and a Race Face Turbine handlebar, making it an affordable yet high-performance option for those looking to save money without sacrificing quality.``
>
> **Reference**
>
> [1] Judging LLM-as-a-Judge with MT-Bench and Chatbot Arena. Zheng et al., NeurIPS 2023 Datasets and Benchmarks Track.
>
> [2] Verbosity Bias in Preference Labeling by Large Language Models. Saito et al., 2023.
>
> [3] Holistic Evaluation for Interleaved Text-and-Image Generation. Liu et al., EMNLP 2024.

---

> ### Author Response · Authors · 2024-11-23
> **Official Responses to Reviewer YfET (Part 2)**
>
> **W2: Attribution of performance improvement**
>
> We have an isolated experiment in Table 2 to show a direct comparison between our MoSS and other PEFT methods (i.e., LoRA and MoE-LoRA). All the PEFT methods are trained on our LeafInstruct dataset, and the only difference among the models is the PERT architecture. From Table 2, using MoSS with the same training set achieves significant improvement, which shows the effectiveness of our proposed MoSS framework.
>
> We clarify that the results for the four middle rows in Table 1 reflect the model's original performance without further training on LeafInstruct. We also have the results for full-parameter fine-tuning using LeafInstruct in Table B:
>
> **Table B: Comparison between full finetuning and parameter-efficient tuning with MoSS.**
>
> | Model | Text Quality | Perceptual Quality | Image Coherence | TIC | Helpfulness |
> |----------------|:--------------:|:--------------------:|:-----------------:|:------:|:-------------:|
> | Full Finetuning| **3.20** | 3.21 | 2.98 | **3.60** | **3.23** |
> | MoSS | 2.61 | **3.62** | **3.41** | 3.54 | 2.71 |
>
>
> The first row represents the results of fully fine-tuning EMU2 on our proposed LeafInstruct dataset. The second row shows the results of fine-tuning EMU2 using our proposed MoSS framework. As observed, while full fine-tuning allows EMU2 to achieve better performance on text generation, the model demonstrates inferior performance on image generation due to its lack of inductive bias. In contrast, tuning with MoSS, which incorporates ConvLoRA, significantly improves image generation performance, even though the number of trained parameters in full fine-tuning is substantially larger than that of MoSS. These results clearly highlight the advantages of integrating ConvLoRA into the transformer architecture for processing visual information.
>
> **W3: Computation cost of ConvLoRA**
>
> We did a comparison of the computational cost between using linear LoRA and our ConvLoRA, respectively. We compute the inference time for generating 1,000 images for each model. The total inference times for linear LoRA and ConvLoRA are 4,380 seconds and 5,910 seconds, respectively. The difference between the two models is around 1.5 seconds per image, indicating the computational cost increased by ConvLoRA is not significant.

---

> > ### Comment · Reviewer_YfET · 2024-11-25
> >
> > Thank you for your thoughtful rebuttal. The additional experiments have effectively addressed my concerns regarding the decline in text quality and the attribution of performance improvements. As a result, I believe the score should be adjusted to 7. However, since the system only allows scores of 6 or 8, I will not make any changes within the system. Nonetheless, I would like to note that my intended score is 7.

---

> > > ### Author Response · Authors · 2024-11-25
> > >
> > > We sincerely appreciate your response, encouragement, and the time and efforts you have dedicated to improving our work. We hope the PCs and ACs can take this into consideration when making the decision.

---

> ### Author Response · Authors · 2024-11-30
>
> Dear Reviewer YfET,
>
> We sincerely appreciate your thoughtful evaluation and your explicit intention to raise the score to 7, acknowledging our responses and additional experiments have effectively addressed your previous concerns.
>
> Given that the system only allows scores of 6 or 8, we would like to respectfully note that the score of 8 (“Accept, good paper”) in ICLR’s scale closely aligns with a score of 7 in other top-tier conferences like NeurIPS, which defines 7 as “Accept: Technically solid paper with high impact on at least one sub-area.” We bring this up in hopes to make sure your score is aligned with your intention and this perspective might be helpful as you finalize your assessment within the system’s constraints.
>
> Thank you again for your careful consideration and detailed feedback throughout this process.
>
> Best regards,
>
> Authors

---

### Official Review · Reviewer_izYW · 2024-11-03

**Soundness:** 3
**Presentation:** 3
**Contribution:** 3
**Rating:** 6
**Confidence:** 3

**Summary:**

This paper proposes a new modality-specialized training method called MOSS and an instruction-turning dataset for interleaved text-and-image generation. By adopting MOSS on two existing frameworks, they show improvements on an interleaved evaluation benchmark (InterleavedBench).

**Strengths:**

1. This paper proposed a novel design that enhances VLGs to generate interleaved content with modality-specialized parameters and adaptation architectures.
2. This paper introduces an open-sourced large-scale instruction-tuning dataset that allows interleaved multi-image and text input and output.

**Weaknesses:**

1. The proposed convolutional LoRA (Equation 4) is similar to the LoRA proposed in [1]. The authors claim that their new LoRA could alleviate the information loss, yet no experimental comparison between the two kinds of Conv LoRAs is provided.
2. The evaluation is limited.
    1. They only evaluate on InterleavedBench and an image editing benchmark called MagicBrush. The coverage of the evaluation is relatively small.
    2. Since the proposed model can do both image and text generation, more benchmarks could be included to measure the model performance from different perspective. E.g., multimodal understanding benchmarks like MMMU, MathVista, VQAv2, POPE. image generation benchmarks such as GenEval and T2I-CompBench.
3. For the proposed data:
    1. The automatic data annotation pipeline  mentioned in Line 298 is not elaborated. How did the authors acquire LeafInstruct from existing academic dataset?
    2. According to Line 299, the details of dataset construction are shown in Table 3. I cannot find any construction details in Table 3.
4. This paper is not well organized and written. Some reference (such as the above table reference) is not accurate. This may cause it is not easy to read and understand.
5. The improvements are relatively limited. Adding MOSS only gets the open-source state-of-the-art performance on 3/5 metric. Adding MOSS onto Chameleon even causes text quality drop. Did the authors have hypnosis or investigate these results?
6. No qualitative results of Chameleon-MOSS.
[1] Convolution meets lora: Parameter efficient finetuning for segment anything model.

**Questions:**

Please see above.

---

> ### Author Response · Authors · 2024-11-23
> **Official Responses to Reviewer izYW (Part 1)**
>
> We thank the reviewer for their constructive comments and valuable insights to improve this work. The responses to your comments are as follows:
>
> **W1: Experimental comparison with two ConvLoRA.**
> To show the benefits of our modified ConvLoRA architecture compared to the ConvLoRA proposed in [3] denoted as SAM-ConvLoRA, we replace the ConvLoRA in MoSS with SAM-ConvLoRA. Specifically, we set the rank of project-down and project-up matrices in SAM-ConvLoRA to 256 which is the same number of ranks in our proposed MoSS-ConvLoRA, and adopt the multi-scale convolution kernels to the size of 2x2 and 4x4.
>
>
> **Table A: Comparison of two types of ConvLoRA.**
>
> | Model | Text Quality | Perceptual Quality | Image Coherence | TIC | Helpfulness |
> |----------------------------------|:------------:|:------------------:|:---------------:|:----:|:-----------:|
> | MoSS w/ MoSS-ConvLoRA (Ours) | **2.61** | **3.62** | **3.41** | **3.54** | **2.71** |
> | MoSS w/ SAM-ConvLoRA | 2.50 | 3.33 | 3.17 | 3.50 | 2.41 |
>
> As shown in Table A, Our MoSS-ConvLoRA consistently outperforms the previous SAM-ConvLoRA on all other evaluation aspects, which demonstrates the superiority of our proposed ConvLoRA architecture. Particularly, our MoSS-ConvLoRA achieves notably better visual qualities, including perceptual quality and image coherence, thanks to our novel design that our convolution operation is applied to the full-rank original image features instead of low-rank image features as in SAM-ConvLoRA.
>
> **W2: Adding more evaluation benchmarks.**
>
> We would like to first clarify that the main focus of our work is interleaved generation. The main evaluation benchmark we used, i.e., InterleavedBench, has already covered a broad array of widely adopted interleaved generation tasks collected from existing well-established academic datasets, including multimodal script generation from WikiHow (Yang et al., 2021), visual storytelling from VIST (Huang et al., 2016), multi-concept image composition from CustomDiffusion (Kumari et al., 2023), activity generation from ActivityNet (Krishna et al., 2017), image editing from MagicBrush (Zhang et al., 2023a), and so on. Given that InterleavedBench is currently the most comprehensive and well-established benchmark for interleaved generation, we believe our evaluation coverage is already comprehensive and sufficient.
>
> Nevertheless, we sincerely appreciate the reviewer’s suggestion to include more benchmarks to measure the performance from different perspectives. We are currently working on these additional experiments and will report the results as soon as possible.
>
> **W3: More details on the proposed dataset.**
>
> Due to the space constraint, we elaborated on the dataset construction details in Appendix B, where we present the detailed process of how we construct LeafInstruct from existing data resources.
>
> In Line 299, “We include the details on dataset construction in Table 3 in Appendix B.”, the reference “in Table 3” is a typo, and the corrected version should be “**We include the details on dataset construction in Appendix B.**” We thank the reviewer for spotting this typo for us and we will correct this in the revision.
>
> **W4: The writing and organization of the paper.**
>
> We thank the reviewer for the suggestion. We will correct the specified reference errors in the revised paper.

---

> ### Author Response · Authors · 2024-11-23
> **Official Responses to Reviewer izYW (Part 2)**
>
> **W5: Limited improvement of MoSS.**
>
> **First**, we would like to justify that our approach can significantly improve the original VLG by a large margin (e.g., 97.76% on the average of 5 aspects), which effectively demonstrates the effectiveness of our approach. It is also worth noting that among the 5 evaluation aspects, ***helpfulness*** is the most important aspect as it holistically measures if the generated content is useful for the task. From Table 1, our Emu2+MoSS model significantly outperformed all the open-sourced VLG baselines on ***helpfulness***, which also shows that our framework can effectively improve the model’s overall capabilities and instruction-following ability.
>
> **Second**, the reason why adding MoSS in Chameleon can cause a slight performance drop in text quality is that the original Chameleon usually generates long and verbose text responses but with no image output. On the contrary, as the text responses in our LeafInstruct dataset are more concise to allow for including more images, after interleaved instruction tuning, our model learns to generate more concise text responses. Specifically, the average generated word length of the original Chameleon is 653, whereas that of Chameleon-MoSS is 166. The verbose responses from the original Chameleon are preferred by the LLM judge due to their verbosity bias [1,2], leading to a slight drop in the text quality of Chameleon-MoSS in LLM-based evaluation.
>
> To better support this analysis, we further conduct a human evaluation of the text quality of the two models by randomly sampling 100 instances from InterleavedBench. We ask a human annotator to select the preferred text responses given the system outputs from two models. We report the Win-Tie-Loss results in Table B. Win means our Chameleon-MoSS is better than the original Chameleon, Tie means the quality of two responses is equally good, and Loss means the original Chameleon is better.
>
> **Table B: Human evaluation results on text quality of the original Chameleon and our Chameleon-MoSS. "Win" indicates our Chameleon-MoSS's responses are preferred by humans.**
>
> | Wins | Ties | Losses |
> |:------:|:------:|:--------:|
> |  28  |  54  |   18   |
>
> From Table B, the text quality of our Chameleon-MoSS is actually better than the original Chameleon. One issue we frequently observed in the original Chameleon is the text responses are overly verbose and sometimes even severely repetitive. In our evaluation protocol of text quality adopted from InterleavedEval [3], such verbosity and repetitiveness are not penalized, making the automatic evaluation results heavily biased towards the longer responses from the original Chameleon. In addition, we also provide a qualitative comparison between the original Chameleon and Chameleon-MOSS, which also shows that applying MoSS will not hurt text quality and can alleviate the repetition issue.
>
> *Example:*
>
> *Instruction:* Describe the new features of Trek's expanded Slash lineup, highlighting the availability of carbon and aluminum models and the introduction of a more affordable top-of-the-line option.
> Input Context: Trek expands its award-winning Slash lineup in Waterloo, WI.
>
> *Original Chameleon Response:* ``The new features include the introduction of the Alpha Platinum Aluminum model, a lower-priced carbon version, and upgrades across all models. The current price range for the new models is $5,000 to $20,000. The features of the new full-carbon Slash 9.9 mountain bike include carbon wheels and Rapid Drive hub. The price range for the new full-carbon Slash 9.9 mountain bike is $20,000 to $25,000. The price range for the new aluminum Slash 9.9 mountain bike is $10,000 to $15,000……``
>
> *Chameleon-MoSS Response (ours):* ``There are several options in Trek's Slash lineup. For example, you can now purchase the new top-of-the-line Slash 9.9 Alpha Platinum Aluminum, which comes equipped with top-notch components such as an SRAM X01 Eagle drivetrain and a Race Face Turbine handlebar, making it an affordable yet high-performance option for those looking to save money without sacrificing quality.``
>
> **W6: Qualitative results of Chameleon-MoSS.**
>
> Thanks for pointing this out. We are preparing the updated version of our paper by considering all the reviewers' comments including adding the qualitative results of Chameleon-MoSS. We will upload the new version of the paper soon.

---

> ### Author Response · Authors · 2024-11-24
> **Official Responses to Reviewer izYW (Part 3)**
>
> **W2: Adding more evaluation benchmarks. (continued)**
>
> We have finished all the additional experiments requested by the reviewer. The detailed implementation details and results are as follows. To show that our MoSS framework can also excel on tasks requiring single modality outputs i.e., the output only contains text or an image, we evaluate its performance on widely adopted image understanding benchmarks including MMBench, MME, MMMU, Pope, and MM-Vet, and text-to-image generation benchmarks including MSCOCO 30K [7], and GenEval [9].
>
> Since LeafInstruct mainly targets tasks with interleaved outputs, we augmented it with 500,000 instances from Vision-Flan [5], a popular visual-instruction tuning dataset targeting image understanding, and 500,000 instances from LAION-COCO [6], a standard training dataset for text-to-image generation. We finetune Emu2 with LoRA, MoE-LoRA, and MoSS on the mixed dataset. We report their performance of multimodal understanding tasks in Table C, and text-to-image generation tasks in Table D.
>
> **Table C: Results on widely adopted multimodal understanding benchmarks.**
>
> | Model | MMBench | MME | MMMU | Pope | MM-Vet |
> |------------|:-------:|:-------:|:------:|:------:|:------:|
> | LoRA | 54.1 | 1148.0 | 33.7 | 87.3 | 31.3 |
> | MoE-LoRA | 54.6 | 1170.3 | 34.1 | **88.1** | 31.9 |
> | MoSS | **56.0** | **1278.4** | **35.8** | 87.6 | **34.1** |
>
> From Table C, our MoSS outperforms previous LoRA and MoE-LoRA on most of the multimodal understanding benchmarks by a notable margin, which demonstrates that MoSS can be well generalized to diverse multimodal comprehension tasks. Note that for all multimodal understanding tasks, we use their official implementation for evaluation.
>
> **Table D: Results on widely adopted text-to-image generation benchmarks. Note that the FID metric is the lower the better.**
>
> | Model | MSCOCO-30K FID (↓) | GenEval (↑) |
> |------------|:--------------:|:-------:|
>  | LoRA | 23.4 | 26.8 |
> | MoE-LoRA | 22.7 | 28.1 |
> | MoSS | **18.2** | **28.9** |
>
>
> For MSCOCO-30K, following the previous evaluation protocol [8], we randomly sample 30,000 captions from the validation set of MSCOCO and generate 30,000 images. We report the FID between the 30,000 generated images and real images from the validation set of MSCOCO (Note for FID, the lower the better). For GenEval, we adopt their official implementation of the evaluation. From Table D, our MoSS achieves better performance on both benchmarks, showing the effectiveness and generalizability of our approach. Notably, our MoSS achieves significantly better FID on MSCOCO-30K, which validates that our ConvLoRA can effectively improve the quality of generated images.
>
>
> Please let us know if our responses and additional experiments have addressed your concerns and if you still have remaining questions. We are looking forward to your reply and sincerely hope you can reconsider your evaluation. Thank you again for your time and feedback in improving this work.
>
> **Reference**
>
> [1] Judging LLM-as-a-Judge with MT-Bench and Chatbot Arena. Zheng et al., NeurIPS 2023 Datasets and Benchmarks Track.
>
> [2] Verbosity Bias in Preference Labeling by Large Language Models. Saito et al., 2023.
>
> [3] Convolution Meets LoRA: Parameter Efficient Finetuning for Segment Anything Model. Zhong et al., 2024
>
> [4] Holistic Evaluation for Interleaved Text-and-Image Generation. Liu et al., EMNLP 2024.
>
> [5] Vision-Flan: Scaling Human-Labeled Tasks in Visual Instruction Tuning. Xu et al., ACL 2024
>
> [6] https://huggingface.co/datasets/laion/laion-coco
>
> [7] Microsoft COCO: Common Objects in Context. Lin., et al.
>
> [8] Generative multimodal models are in-context learners. Sun., et al., CVPR 2024
>
> [9] GenEval: An Object-Focused Framework for Evaluating Text-to-Image Alignment. Ghosh et al., CVPR 2023.

---

> > ### Comment · Reviewer_izYW · 2024-11-25
> >
> > Thanks for providing more evaluation results and detailed explanations. However, the added comparison is between MOSS and its ablation variants. Could you provide comparisons with other models such as Chameleon and GILL?

---

> ### Author Response · Authors · 2024-11-25
>
> Thank you for your response. We added the results of Chameleon on the additional evaluation benchmarks in Table E. From the results, our MoSS (based on Emu2) outperforms the original Chameleon baseline on 5 out of 7 benchmarks by a significant margin, demonstrating the strong capabilities and generalizability of our approach.
>
> **Table E: Comparison between Chameleon and our MoSS on additional evaluation benchmarks.**
>
> | Model      | MMBench | MME     | MMMU   | Pope   | MM-Vet | MSCOCO (↓) | GenEval |
> |------------|:-------:|:-------:|:------:|:------:|:------:|:----------:|:-------:|
> | Chameleon  | 32.7    | 604.5   | **38.8** | 59.8   | 9.7    | 26.7       | **39.0** |
> | MoSS (Ours)       | **56.0** | **1278.4** | 35.8   | **87.6** | **34.1** | **18.2** | 28.9    |
>
>
> For GILL, we found it very difficult to generalize to many of your requested benchmarks, as GILL is highly specialized in text-and-image generation tasks without considering much multimodal understanding capabilities. For example, GILL is solely trained on Conceptual Captions (CC3M) to generate interleaved images and captions, but it lacks training on multimodal comprehension or instruction following data. In addition, Chameleon (released in May 2024) is a more recent and advanced model compared with GILL (released in May 2023). Therefore, we believe comparing Chameleon with our model is sufficient to demonstrate the effectiveness and superiority of our approach. Given the time costs of these additional experiments are high and the remaining time of the discussion period is limited, we will include more baselines on these additional benchmarks for reference in the final version.
>
> Please let us know if this fully addresses your question and if you still have any remaining concerns. Thank you again for your time and engagement in the discussion.

---

> > ### Comment · Reviewer_izYW · 2024-11-25
> >
> > Thanks for your response. Most of the questions I raised have been well resolved, and I have updated the rating accordingly.

---

> > > ### Author Response · Authors · 2024-11-25
> > >
> > > Thank you very much for your positive feedback. We sincerely appreciate your updated evaluation and recognition of our work.

---

### Official Review · Reviewer_pC3r · 2024-11-03

**Soundness:** 3
**Presentation:** 3
**Contribution:** 3
**Rating:** 5
**Confidence:** 4

**Summary:**

This paper aims to improve the understanding ability of existing Vision-Language Generalists (VLGs) and propose the Modality-Specialized Synergizers (MoSS). Moss focuses on two things: 1) To build high-quality interleaved text and images, Moss modifies the connectors (such as the Linear layers between the image encoder and LLMs) in VLGs) by introducing a linear LoRA (for textual inputs) and Convolutional LoRA (For image inputs). 2) To improve the understanding of perform interleaved tasks. MoSS develops a high-quality instruction dataset (with 184,982 instances). Experiment results show the improvements of the introduced two LoRAs and ablations show the efficiency of the convolutional LoRA.

**Strengths:**

1) This paper is written well with clear figures and tables, which make the readers easy to follow the story.

2) The ideas that utilize model-specific adapter to process the text and image make sense to me. Such adapter may capture inherent semantics of corresponding inputs.

3) This paper develops a high-quality interleaved instruction dataset, which will benefit to the VLG community.

4) Experiments and ablations show the improvements and efficiency of the proposed modules

**Weaknesses:**

1) One of the core concerns is the novelty of the two LoRA types. Given the facts that both Linear LoRA and convolutional LoRA are not new to recent vision-language models. I think the contributions are limited.

2) Technically, MoSS beliefs that the linear layer could lose the visual information and adopts the convolution LoRA to capture local patch features. However, the ViT-based image encoder in EMU (ViT are often used in VLMs) already flatten an image into a token sequence and employs self-attention  to model their dependents. That is that the ViT outputs inherently contain the local information, and the linear adapter act as a semantic translator that map the visual features into LLM space. Thus, I do not think the convolutional LoRA is necessary for VLG, mathmetically.

3) MoSS argues that previous methods use the same framework for both text and image. If we think the model-specific encoder and the shared adapter as a whole module, we find the they use different frameworks to process the text and image.

4) Can the authors apply MoSS to more powerful models, such as Emu 3 to test it robustness?

**Questions:**

Please see the Weaknesses section.

---

> ### Author Response · Authors · 2024-11-23
> **Official Responses to Reviewer pC3r**
>
> We thank Reviewer pC3r for the constructive comments and valuable insights to improve this work. The responses to the comments are as follows:
>
> **W1: Novelty of two LoRA types.**
>
> We would like to clarify and justify the key novelty and contributions of this work as follows:
> Our work is the first to integrate convolution LoRA into autoregressive generative models backed by large language models (LLMs). This requires new design strategies to reconcile the discrepancy between the spatial operation in convolution and the sequential autoregressive generation process of language models. For example, during inference, we introduce an on-the-fly partial convolutional mechanism tailored for autoregressive generation (as shown in the right part of Figure 2), where the convolution kernel operated on each image patch only covers its neighboring patches on the left and top, as the patches on the right and bottom have not been generated yet. In addition, while previous works such as mixture-of-LoRA [3,4] propose to leverage separate linear LoRAs with the same architecture for different tasks or different modalities, we are the first to integrate two distinct LoRA architectures within a single model for processing images and text, respectively. We will add more detailed explanations and discussion regarding the contributions in the revised version.
>
> **W2: The necessity of convolutional LoRA.**
>
> Several recent studies [1,2] have demonstrated that a plain ViT-based image encoder lacks vision-specific inductive bias for dense predictions. Specifically, the quantitative analysis of [5,6] proves that multi-head self-attentions are more capable of modeling global shapes and structures of a scene in an image, while convolution layers are more capable of capturing local information such as edges and textures due to their strong inductive bias such as the local spatial operation. Through the lens of Fourier analysis [1,5,6], i.e., analyzing the amplitude of Fourier-transformed image features generated by either purely transformer-based vision models or convolution-based vision models, previous studies show that transformer-based models reduce the high-frequency signals in images, acting as a low-pass filter, and conversely, convolution-based models amplify high-frequency components, acting as a high-pass filter. Thus, relying solely on multi-head self-attentions can cause the VLGs to miss important local visual information,  and two structures can be combined to capture richer information from images. In our work, we harmonize two structures by integrating convolutional LoRA into the multi-head self-attention layers for effectively modeling both global and local dependency of image features in image generation. In Table 2, we also empirically proved that applying convolutional LoRA significantly outperforms the baselines using a mixture of linear LoRA, which further justified the necessity of convolutional LoRA.
>
> **W3: Clarification for “same framework for both text and image”.**
>
> We would like to clarify that while VLGs typically use a separate image encoder to convert images into continuous vectors or discrete image tokens, by “same framework” we mean existing methods using the identical architecture within the LLM component of VLGs for processing both images and text. Given that the LLM is responsible for reasoning over multimodal inputs and generating multimodal outputs, we argue that it should have dedicated parameters to mitigate modality conflicts. The effectiveness and necessity of using separate sets of LoRA within LLMs have been demonstrated in prior studies [3,4] in the multimodal domain.
>
> **W4: Applying MoSS to more powerful VLGs.**
>
> Emu2 and Chameleon were two state-of-the-art VLGs at the time of our paper submission. More advanced models, such as Emu3, were published after the abstract deadline and are therefore concurrent with our work. Given that Emu3 employs transformer layers as its building blocks and uses discrete tokens to represent images, which are similar to Chameleon, we argue that our method can be similarly adapted to Emu3. Given the limited time and high demand for computational resources, we will not be able to get the results during the rebuttal period but will include the results in the final version.
>
> **Reference**
>
> [1] Convolution Meets LoRA: Parameter Efficient Finetuning for Segment Anything Model. Zhong et al., ICLR 2024.
>
> [2] Vision Transformer Adapter for Dense Predictions. Chen et al., ICLR 2023.
>
> [3] Mixture of Cluster-conditional LoRA Experts for Vision-language Instruction Tuning. Gou et al., 2023.
>
> [4] Multimodal Instruction Tuning with Conditional Mixture of LoRA. Shen et al., ACL 2024.
>
> [5] Inception Transformer. Si et al., NeurIPS 2022.
>
> [6] How Do Vision Transformers Work? Park et al., ICLR 2022.

---

> ### Author Response · Authors · 2024-11-25
>
> Dear Reviewer pC3r,
>
> We sincerely appreciate the time and effort you've devoted to reviewing our work. We understand that your schedule may be quite busy, and we are truly grateful for your valuable feedback. As the Author-Reviewer discussion phase is ending soon, we would greatly value the opportunity to engage in further discussion with you. Our aim is to gain insights into whether our responses effectively address your concerns and to ascertain if there are any additional questions or points you would like to discuss.
>
> We look forward to the opportunity for further discussion with you. Thank you for your thoughtful consideration.
>
> Best regards,\
> Authors

---

> > ### Comment · Reviewer_pC3r · 2024-11-26
> >
> > I thank the authors for their response. I decided to keep my rating as 5.

---

> ### Author Response · Authors · 2024-11-26
> **Thank you for the prompt response**
>
> We would like to extend our sincerest gratitude for the time and effort you have made in reviewing and providing valuable feedback that is crucial for improving our work.
>
> We greatly value each comment and suggestion from your review. As there is still time remaining before the conclusion of the discussion phase, we would be truly grateful if you could let us know whether there remain any unresolved concerns in your view. In our continuous effort to enhance the quality and impact of our research, we are glad to address any remaining issues during the discussion phase. Thank you a lot!

---

### Author Response · Authors · 2024-12-03
**General Responses**

Dear ICLR PCs, ACs, and all reviewers,

We would like to express our genuine gratitude for your time and efforts in facilitating the discussion regarding our paper. We sincerely appreciate the insightful comments and the recognition of the contributions and quality of our work, including the **novelty of our approach** (Reviewer izYW, YfET, and ccUt), **our curated dataset is high-quality and beneficial** (Reviewer pC3r, izYW, and ccUt), **our solid experiments** showing promising improvement (Reviewer pC3r, YfET, and ccUt), and **clear presentation** (Reviewer pC3r).

We are particularly grateful that Reviewer izYW has increased their score to 6, Reviewer YfET has increased their score to 7, and Reviewer ccUt has maintained their positive assessment. Although we understand that Reviewer pC3r has not engaged in subsequent discussions due to the busy schedule, we believe that our responses have fully addressed most concerns of all the reviewers through clear explanations and additional experiments.

As the discussion is coming to an end, we would like to provide a brief summary of the key points that have been discussed and addressed:
- We have provided a detailed explanation addressing the concerns raised by Reviewer izYW and Reviewer YfET regarding the potential performance drop in text quality when integrating MoSS with Chameleon. To further substantiate our claims, we conducted additional human evaluations. The results indicate that the text outputs generated by Chameleon integrated with MoSS exhibit a better alignment with human preferences compared to the original Chameleon model.
- In response to Reviewer izYW's suggestion, we have included additional results on a diverse set of widely adopted evaluation benchmarks for multimodal understanding and text-to-image generation. These results demonstrate the effectiveness and generalizability of our proposed method across various tasks and datasets.
- We have made substantial clarifications on the novelty, motivations, and technical details as suggested by Reviewer pC3r and Reviewer ccUt. For example, we refer to the theoretical analysis in previous works, i.e., Fourier analysis, to justify the necessity to incorporate convolutional operations in pure transformer-based language models to capture more visual-specific inductive biases for interleaved generation. These revisions aim to better highlight the unique contributions and underlying rationale of our work.

We would like to emphasize the contributions of our work, which have been acknowledged by the reviewers and are important to the community:
- **Novel modality-specialized design:** we introduce MoSS, integrating Convolutional LoRA for images and Linear LoRA for text, effectively capturing modality-specific inductive biases in VLGs.
- **High-quality dataset:** we curate the first large-scale, open-source interleaved instruction tuning dataset with 184,982 instances, providing valuable resources for multimodal generation.
- **State-of-the-art performance:** we demonstrate significant improvements in interleaved text-image generation tasks and instruction-following capabilities across diverse benchmarks.
- **Strong generalizability and robustness to different VLGs:** we validate MoSS on multiple VLG backbones, proving its generalizability to both discrete and continuous image token spaces.

Finally, we deeply value the constructive comments provided by the reviewers. In response, we have carefully revised our paper based on the feedback received. Considering the contributions made, we hope our work can provide new insights and valuable resources to the multimodal and broader communities, and contribute to their further development.

Sincerely,

Authors

---

### Meta-Review · Area_Chair_eohS · 2024-12-18

**Metareview:**

This paper introduces Modality-Specialized Synergizers to improve interleaved text-image generation in Vision-Language Generalists (VLGs). By integrating modality-specific LoRAs and providing a large-scale interleaved instruction-tuning dataset (LeafInstruct), the authors show meaningful gains in multimodal tasks. The reviewers agreed that the paper is clearly written, the dataset is beneficial, and the method improves on strong baselines. While some raised questions regarding novelty (as LoRA variants are known) and why convolution is needed given ViT representations, the authors clarified the theoretical and empirical motivations. Although certain concerns about performance fluctuations and incomplete comparisons persist, the open-source dataset, and the constructive revisions merit recognition. Given the positive feedbacks, I recommend acceptance.

**Additional Comments On Reviewer Discussion:**

During the discussion, the authors provided extra experiments and clarifications that addressed key concerns about novelty, dataset construction, and performance drops. Their engagement led some reviewers to upgrade their evaluations. While not all reviewers revisited their scores, the consensus moved toward recognizing the proposed value. The added benchmarks and analyses strengthen the paper's case.

---

### Decision · Program_Chairs · 2025-01-22

Accept (Poster)